# Concept Gradients: Concept-based Interpretation Without Linear Assumption

**Andrew Bai**
Department of Computer Science
University of California, Los Angeles
`andrewbai@cs.ucla.edu`

**Chih-Kuan Yeh**
Department of Computer Science
Carnegie Mellon University
`cjyeh@cs.cmu.edu`

**Neil Y. C. Lin**
Department of Bioengineering
University of California, Los Angeles
`neillin@g.ucla.edu`

**Pradeep Ravikumar**
Department of Computer Science
Carnegie Mellon University
`pradeepr@cs.cmu.edu`

**Cho-Jui Hsieh**
Department of Computer Science
University of California, Los Angeles
`chohsieh@cs.ucla.edu`

## Abstract

Concept-based interpretations of black-box models are often more intuitive than feature-based counterparts for humans to understand. The most widely adopted approach for concept-based gradient interpretation is Concept Activation Vector (CAV). CAV relies on learning linear relations between some latent representations of a given model and concepts. The premise of meaningful concepts lying in a linear subspace of model layers is usually implicitly assumed but does not hold true in general. In this work we proposed Concept Gradients (CG), which extends concept-based gradient interpretation methods to non-linear concept functions. We showed that for a general (potentially non-linear) concept, we can mathematically measure how a small change of concept affects the model's prediction, which is an extension of gradient-based interpretation to the concept space. We demonstrate empirically that CG outperforms CAV in evaluating concept importance on real world datasets and perform a case study on a medical dataset. The code is available at `github.com/jybai/concept-gradients`.

## 1 Introduction

Explaining the prediction mechanism of machine learning models is important, not only for debugging and gaining trust, but also for humans to learn and actively interact with them. Many feature attribution methods have been developed to attribute importance to input features for the prediction of a model (Sundararajan et al., 2017; Zeiler & Fergus, 2014). However, input feature attribution may not be ideal in the case where the input features themselves are not intuitive for humans to understand. It is then desirable to generate explanations with human-understandable concepts instead, motivating the need for **concept-based explanation**. For instance, to understand a machine learning model that classifies bird images into fine-grained species, attributing importance to high-level concepts such as body color and wing shape explains the predictions better than input features of raw pixel values (see Figure 1).

The most popular approach for concept-based interpretation is Concept Activation Vector (CAV) Kim et al. (2018). CAV represents a concept with a vector in some layer of the target model and evaluates the sensitivity of the target model's gradient in the concept vector's direction. Many followup works are based on CAV and share the same fundamental assumption that concepts can be represented as a linear function in some layer of the target model (Ghorbani et al., 2019; Schrouff et al., 2021). This assumption generally does not hold, however, and it limits the application of

Black footed Albatross

| Rank | Concepts | CG score |
|---|---|---|
| 1 | has_size::medium_(9_-_16_in) | 2.134 |
| 2 | has_bill_length::about_the_same_as_head | 1.905 |
| 3 | has_bill_shape::hooked_seabird | 1.834 |
| 4 | has_wing_pattern::solid | 1.825 |
| 5 | has_wing_shape::pointed-wings | 1.609 |
| 6 | has_shape::duck-like | 1.443 |
| 7 | has_under_tail_color::grey | 1.419 |
| 8 | has_primary_color::grey | 1.404 |
| 9 | has_forehead_color::white | 1.123 |
| 10 | has_upper_tail_color::grey | 1.082 |

Figure 1: Comparison of feature-based interpretation heatmap (left: Integrated Gradients) and concept-based importance score (right: Concept Gradients) for the model prediction of "Black footed Albatross". Attribution to high-level concepts is more informative to humans than raw pixels.

concept-based attribution to relatively simple concepts. Another problem is the CAV concept importance score. The score is defined as the inner product of CAV and the input gradient of the target model. Inner product captures correlation, not causation, but the concept importance scores are often perceived causally and understood as an explanation for the predictions it does.

In this paper, we rethink the problem of concept-based explanation and tackle the two weak points of CAV. We relax the linear assumption by modeling concepts with more complex, non-linear functions (e.g. neural networks). To solve the causation problem, we extend the idea of taking gradients from feature-based interpretation. Gradient-based feature interpretation assigns importance to input features by estimating input gradients, i.e., taking the derivative of model output with respect to input features. Input features corresponding to larger input gradients are considered more important. A question naturally arises: is it possible to extend the notion of input gradients to "concept" gradients? Can we take the derivative of model output with respect to post-hoc concepts when the model is not explicitly trained to take concepts as inputs?

We answer this question in affirmative and formulate Concept Gradients (CG). CG measures how small changes of concept affect the model's prediction mathematically. Given any target function and (potentially non-linear) concept function, CG first computes the input gradients of both functions. CG then combines the two input gradients with the chain rule to imitate taking the derivative of the target with respect to concept through the shared input. The idea is to capture how the target function changes locally according to the concept. If there exists a unique function that maps the concept to the target function output, CG exactly recovers the gradient of that function. If the mapping is not unique, CG captures the gradient of the mapping function with the minimum gradient norm, which avoids overestimating concept importance. We discover that when the concept function is linear, CG recovers CAV (with a slightly different scaling factor), which explains why CAV works well in linearly separable cases. We showed in real world datasets that the linear separability assumption of CAV does not always hold and CG consistently outperforms CAV. The average best local recall@30 of CG is higher than the best of CAV by 7.9%, while the global recall@30 is higher by 21.7%.

## 2 PRELIMINARIES

**Problem definition** In this paper we use $f : \mathbb{R}^d \to \mathbb{R}^k$ to denote the machine learning model to be explained, $x \in \mathbb{R}^d$ to denote input, and $y \in \mathbb{R}^k$ to denote the label. For an input sample $\hat{x} \in \mathbb{R}^d$, concept-based explanation aims to explain the prediction $f(\hat{x})$ based on a set of $m$ concepts $\{c_1, \ldots, c_m\}$. In particular, the goal is to reveal how important is each concept to the prediction. Concepts can be given in different ways, but in the most general forms, we can consider concepts as functions mapping from the input space to the concept space, denoted as $g : \mathbb{R}^d \to \mathbb{R}$. The function can be explicit or implicit. For example, morphological concepts such as cell perimeter and circularity can be given as explicit functions defined for explaining lung nodule prediction models. On the other hand, many concept-based explanations in the literature consider concepts given as a set of examples, which are finite observations from the underlying concept function. We further assume

$f$ and $g$ are differentiable which follows the standard assumption of gradient-based explanation methods.

We define a local concept relevance score $R(\hat{x}; f, g)$ to represent how concept $\hat{c}$ affects target class prediction $\hat{y} = f(\hat{x})$ on this particular sample $\hat{x}$. Local relevance is useful when analyzing individual data samples (e.g. explain what factors lead to a bank loan application being denied, which may be different from application to application). Further, by aggregating over all the input instances of class $y$, $Z_y = \{(x_1, y), \ldots, (x_n, y)\}$, we can define a global concept relevance score $R(Z_y; f, g)$ to represent overall how a concept affects the target class prediction $y$. Global relevance can be utilized to grasp an overview of what the model considers importance (e.g. good credit score increases bank loan approval). The goal is to calculate concept relevance scores such that the scores reflect the true underlying concept importance, possibly aligned with human intuition.

**Recap of Concept Activation Vector (CAV)**   Concept activation vector is a concept-based interpretation method proposed by Kim et al. (2018). The idea is to represent a concept with a vector and evaluate the alignment between the input gradients of the target model and the vector. In order for the concept to be represented well by a vector, the concept labels must be linearly separable in the vector space. The authors implicitly assumed that there exists a layer in the target model where concept labels can be linearly separated.

Let $\boldsymbol{v}_c$ denote the concept activation vector associated with concept $c$. The authors define the conceptual sensitivity score to measure local concept relevance

$$R_{\text{CAV}}(x; f, \boldsymbol{v}_c) := \nabla f(x) \cdot (\boldsymbol{v}_c / \|\boldsymbol{v}_c\|). \tag{1}$$

If we view CAV as a linear concept function, $R_{\text{CAV}}$ is the (normalized) inner product of the gradients of the target and concept function. The main caveat with the CAV conceptual sensitivity scores is that the underlying concept function is not guaranteed to lie in the linear subspace of some neural activation space. Attempting to fit the concept function with a linear model likely leads to poor results for non-trivial concepts, leading to inaccurate conceptual sensitivity scores.

**GC: extending CAV to non-linear concepts**   Let us consider modeling concepts with general, non-linear function $g$ instead to relax the assumption on linear separability. We define the non-linear generalization of inner product concept-based interpretation, Gradient Correlation (GC), as follows

$$R_{\text{GC}}(x; f, \boldsymbol{v}_c) := \nabla f(x) \cdot (\nabla g(x) / \|\nabla g(x)\|). \tag{2}$$

GC takes the gradients from the target and concept function and calculates the linear correlation of gradients in each dimension in the shared input feature space. Intuitively, if gradients magnitudes coincide in similar dimensions (large inner product), then the concept is relevant to the target function. CAV is a special case of GC where $g$ is limited to a linear function.

The main caveat of GC is the inner product of gradients between the target and concept only yields correlation. In some applications correlation is already sufficient for interpretation, which explains the success of CAV. However, one may be tempted to ask: if a mapping from the concept to the target function output exists, can we retrieve the gradient of the target function output with respect to concepts to evaluate "causal" interpretation? This is an extension of typical input gradient approaches for feature-based interpretations (Ancona et al., 2017; Shrikumar et al., 2017; Sundararajan et al., 2017). Is it possible to calculate this concept gradient given only access to the gradients $\nabla f(x)$ and $\nabla g(x)$? The answer lies in the shared input feature space of $f$ and $g$ where gradients can be propagated with chain rule.

## 3   PROPOSED METHOD

### 3.1   DEFINITION OF CONCEPT GRADIENTS (CG)

We define the **Concept Gradients** (CG) to measure how small perturbations on a concept $g$ affect the target function output $f$ through gradients:

$$R_{\text{CG}}(x; f, g) := \nabla g(x)^{\dagger} \nabla f(x) = \frac{\nabla g(x)^T}{\|\nabla g(x)\|^2} \cdot \nabla f(x) \tag{3}$$

where $\nabla g(x)^\dagger$ is the Moore–Penrose inverse (pseudo-inverse) of $\nabla g(x)$. Pseudo-inverse is a generalized version of matrix inversion—when the inverse does not exist, it forms the (unique) inverse mapping from the column space to the row space of the original matrix, while leaving all the other spaces untouched. Using pseudo-inverse prevents mis-attribution of importance from other spaces irrelevant to $\nabla g(x)$. Conveniently, the pseudo-inverse of $\nabla g(x)$ is just its normalized transpose since $\nabla g(x)$ is a $d$-dimensional vector. Suppose a function $h$ that maps the concept $c$ to target function uniquely exists,

$$f(x) = h(c) = h(g(x))$$

then CG exactly recovers the derivative of $h$ which is the importance attribution of the target function with respect to concept

$$h'(c) = \frac{\mathrm{d}h(c)}{\mathrm{d}c} = \frac{\mathrm{d}h(g(x))}{\mathrm{d}g(x)} = R_{\mathrm{CG}}(x; f, g).$$

Let us illustrate the intuition CG with a simplified case where $x$, $y$ and $c$ are all scalars ($k = d = 1$). Our goal is to represent $y$ as a function of $c$ to obtain the derivative $\frac{\mathrm{d}y}{\mathrm{d}c}$. We can expand the expression with chain rule:

$$\frac{\mathrm{d}y}{\mathrm{d}c} = \frac{\mathrm{d}y}{\mathrm{d}x} \cdot \frac{\mathrm{d}x}{\mathrm{d}c} = f'(x) \cdot \frac{\mathrm{d}x}{\mathrm{d}c}$$

The derivative $\frac{\mathrm{d}x}{\mathrm{d}c}$ is the remaining term to resolve. Notice that since $x$ and $c$ are scalars, $\frac{\mathrm{d}x}{\mathrm{d}c}$ is simply the inverse of $\frac{\mathrm{d}c}{\mathrm{d}x} = g'(x)$ assuming $\frac{\mathrm{d}c}{\mathrm{d}x} \neq 0$. Thus, concept gradients exactly recovers $\frac{\mathrm{d}y}{\mathrm{d}c}$:

$$\frac{\mathrm{d}y}{\mathrm{d}c} = f'(x) \cdot \left(\frac{\mathrm{d}c}{\mathrm{d}x}\right)^{-1} = f'(x) \cdot g'(x)^{-1} = f'(x) \cdot g'(x)^\dagger = R_{\mathrm{CG}}(x). \tag{4}$$

The pseudo-inverse in (3) extends this derivation to the general case when $x$, $y$, and $c$ have arbitrary dimensions. Details can be found in Appendix B.

## 3.2 IMPLEMENTATION OF CG

Implementing CG is rather simple in practice. Given a target model $f$ to interpret, CG can be calculated if the concept model $g$ is also provided by the user. When concepts are given in the form of positive and negative samples by users, instead of an explicit concept function, we can learn the concept function from the given samples. Generally, any concept model $g$ sharing the same input representation as $f$ would suffice. Empirically, we discovered that the more similar $f$ and $g$ is (in terms of both model architecture and weight), the better the attribution performance. Our hypothesis is since there might be redundant information in the input representation to perform the target prediction, many solutions exist (i.e. many $f$ can perform the prediction equally well). The more similar $g$ utilizes the input information as $f$, the more aligned the propagation of gradient through the share input representation is.

We propose a simple and straightforward strategy for training concept models $g$ similar to $f$. We train $g$ to predict concepts by finetuning from the pretrained model $f$. The weight initialization from $f$ allows the final converged solution of $g$ to be closer to $f$. We can further constrain $g$ by freezing certain layers during finetuning. The similarity between $f$ and $g$ leads to similar utilization of input representation, which benefits importance attribution via gradients. More details regarding the importance of the similarity between $f$ and $g$ can be found in Section 3.3.

CG can be used to evaluate per-sample (local) and per-class (global) concept relevance score. Following TCAV (Kim et al., 2018), we evaluated global CG by relevance score calculating the proportion of positive local CG relevance over a set of same-class samples $\mathcal{Z}_y = \{(x_1, y), \ldots (x_n, y)\}$.

$$R_{\mathrm{CG}}(\mathcal{Z}_y; f, g) := \frac{|\{(x, y) \in \mathcal{Z}_y : R_{\mathrm{CG}}(x; f, g) > 0\}|}{|\mathcal{Z}_y|}$$

Global CG relevance score can also be aggregated differently, suitable to the specific use case.

## 3.3 SELECTING LAYER FOR ATTRIBUTION

Similar to CAV, CG can be computed at any layer of a neural network by setting $x$ as the hidden state of neurons in a particular layer. The representation of input $x$ is relevant to attribution, as the

information contained differs between representations. Both methods face the challenge of properly selecting a layer to perform calculation.

For a feed-forward neural network model, information irrelevant to the target task is removed when propagating through the layers (i.e. feature extraction). Let us denote the representation of $x$ in the $l^{\text{th}}$ layer of the target model $f$ as $x_{f_l}$. We hypothesized that the optimal layer $l^*$ for performing CG is where the representation $x_{f_{l*}}$ contains **minimally necessary and sufficient information** to predict concept labels. Intuitively, the representation of $x_{f_l}$ needs to contain sufficient information to correctly predict concepts to ensure the concept gradients $\nabla g(x)$ are accurate. On the other hand, if there is redundant information in $x_{f_l}$ that can be utilized to predict the target $y$ and concept $c$, then $g$ may not rely on the same information as $f$, which causes misalignment in gradients $\nabla g(x)$ and $\nabla f(x)$ leading to underestimation of concept importance.

The algorithm for selecting the optimal layer to perform CG is simple. The model $g$ is initialized with weights from $f$ and all the weights are initially frozen. Starting from the last layer, we unfreeze the layer weights and finetune $g$ to predict concepts. We train until the model converges and evaluate the concept prediction accuracy on a holdout validation set. The next step is to unfreeze the previous layer and repeat the whole process until the concept prediction accuracy saturates and no longer improves as more layers are unfrozen. We have then found the optimal layer for CG as well as the concept model $g$.

### 3.4 CONNECTIONS BETWEEN CAV, GC, AND CG

In the special case when $g(x) = v_C \cdot x$ is a linear function (the assumption of CAV), we have $R_{\text{CG}}(x) = v_C^T \nabla f(x)/\|v_C\|^2$. This is almost identical to conceptual sensitivity score in Eq 1 except a slightly different normalization term where CAV normalizes the inner product by $1/\|v_C\|$. Furthermore, the sign of CG and CAV will be identical which explains why CAV is capable of retrieving important concepts under the linearly separable case.

Here we use a simple example to demonstrate that the normalization term could be important in some special cases. Consider $f$ as the following network with two-dimensional input $[x_0, x_1]$:

$$y = 0.1z_0 + z_1, \begin{bmatrix} z_0 \\ z_1 \end{bmatrix} = \begin{bmatrix} 100 & 0 \\ 0 & 1 \end{bmatrix} \begin{bmatrix} h_0 \\ h_1 \end{bmatrix}, \begin{bmatrix} h_0 \\ h_1 \end{bmatrix} = \begin{bmatrix} 0.01 & 0 \\ 0 & 1 \end{bmatrix} \begin{bmatrix} x_0 \\ x_1 \end{bmatrix}, \tag{5}$$

and $c_0 = x_0$, $c_1 = x_1$. Then we know since $y = 0.1z_0 + z_1 = 0.1x_0 + x_1$, the contribution of $c_1$ should be 10 times larger than $c_0$. In fact, $\frac{dy}{dc_0} = 0.1$, $\frac{dy}{dc_1} = 1$ and it's easy to verify that CG will correctly obtain the gradient no matter which layer is chosen for computing (3). However, the results will be wrong when a different normalization term is used when computing concept explanation on the hidden layer $h$. Since $c_0 = 100h_0$, $c_1 = h_1$, $y = 10h_0 + h_1$, we have

For concept $c_0$: $v = \dfrac{dc_0}{dh} = [100, 0]^T$, $u = \dfrac{dy}{dh} = [10, 1]$, $v^T u/\|v\| = 10$, $v^T u/\|v\|^2 = 0.1$

For concept $c_1$: $v = \dfrac{dc_1}{dh} = [0, 1]^T$, $\dfrac{dy}{dh} = [10, 1]$, $v^T u/\|v\| = 1$, $v^T u/\|v\|^2 = 1$. $\tag{6}$

Therefore, the normalization term used in CAV (in red color) will lead to a conclusion that $c_0 > c_1$, while CG (in blue color) will correctly get the actual gradient and conclude $c_1 > c_0$. This is mainly because CG is formally derived from the actual gradient, as we will discuss below. In contrast, CAV is based on the intuition of correlation in the form of gradient inner product and not the exact chain-rule gradient, so its attribution is subject to per-dimension scaling.

Although the normalization term can be important in some special cases, in practice we do not find the attribution results to be much different with different normalization terms in empirical studies. We compared different methods of calculating CG (including different normalization schemes) empirically in Section 4.1. We hypothesis such extreme per-dimensional scaling in input features are less common in well-trained neural networks. Thus, in practice if the concept can be accurately modeled by a linear function, CAV might be a good approximation for concept gradients. When linear separability assumption does not hold, GC might be a good approximation for concept gradients. But in general only CG recovers the concept gradients when feature scaling conditions are not ideal.

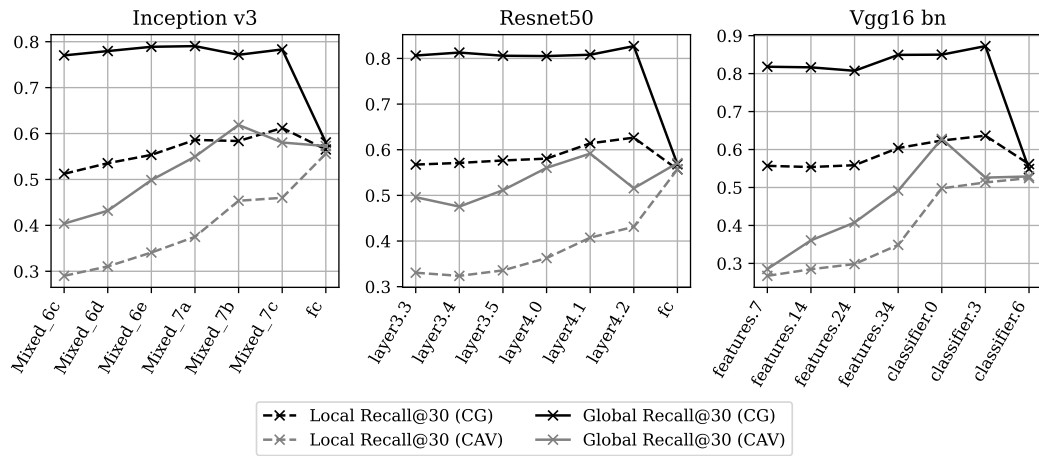

Figure 2: CUB concept recalls for different input representations in various layers and architectures (left to right, deep to shallow layers). CG consistently performs better than CAV locally and globally.

# 4 EXPERIMENTAL RESULTS

## 4.1 QUANTITATIVE ANALYSIS

In this experiment, our goal is to quantitatively benchmark how well CG is capable of correctly retrieving relevant concepts in a setting where the ground truth concept importance is available. In this case, the ground truth concept importance consists of human annotations of how relevant a concept is to the model prediction. We can evaluate the quality of concept importance attribution by treating the task as a retrieval problem. Specifically given a data sample and its target class prediction, the concept importance given by CG is used to retrieve the most relevant concepts to the prediction. A good importance attribution method should assign highest concept importance to concepts relevant to the class.

We benchmarked local (per-sample) and global (per-class) concept importance attribution separately. Given an input sample $(x_i, y_i)$, its ground truth relevant concept set $\hat{C}_i$ and the top $k$ attributed local important concept set $C_i$, the local recall@k is defined as $\frac{|\hat{C}_i \cap C_i|}{k}$.

Given a set of data samples from the same class $\mathcal{Z}_y = \{(x_1, y), \ldots, (x_n, y)\}$ and their ground truth relevant concept sets $\{\hat{C}_1, \ldots, \hat{C}_n\}$, we can obtain the class-wise ground truth concept set $\hat{C}_y$ by majority voting. We can also obtain the top $k$ attributed global important concept set $C_y$. The global recall@k is then defined as $\frac{|\hat{C}_y \cap C_y|}{k}$.

The local attribution performance is measured by the average local recall@k (over samples) on a holdout testing set. The global attribution performance is measured by the average global recall@k (over classes) on a holdout testing set.

**Dataset.** We experimented on CUB-200-2011 (Wah et al., 2011), a dataset for fine-grained bird image classification. It consists of 11k bird images, 200 bird classes, and 312 binary attributes. These attributes are descriptors of bird parts (e.g. bill shape, breast pattern, eye color) that can be used for classification. We followed experimental setting and preprocessing in (Koh et al., 2020) where attributes with few samples are filtered out leaving 112 attributes as concepts for interpretation.

**Evaluation.** For CG, we finetuned the target model $f$ for fine-grained bird classification on the concept labels to obtain the concept model $g$. We also evaluated CAV on the corresponding model layers for comparison. All the details regarding model training for reproducibility can be found in Appendix J.2. The local and global recalls for all models and layers are plotted in Fig 2. The layers in the x-axis are shallow to deep from left to right.

Tables 1 and 2 compares the best result of CAV and CG of the three different model architectures. CG consistently outperforms CAV in concept importance attribution (recall) since the non-linearity in CG captures concepts better (higher concept accuracy).

Table 1: Local concept importance attribution comparison on CUB

| Model | Method | Concept Accuracy | Local (per-sample) | | |
|---|---|---|---|---|---|
| | | | R@30 | R@40 | R@50 |
| Inception v3 | CAV | 0.709 | 0.556 | 0.664 | 0.745 |
| | local-CG | 0.791 | 0.612 | 0.718 | 0.790 |
| Resnet50 | CAV | 0.727 | 0.557 | 0.663 | 0.740 |
| | local-CG | 0.793 | 0.627 | 0.726 | 0.794 |
| Vgg16 bn | CAV | 0.703 | 0.525 | 0.628 | 0.708 |
| | local-CG | 0.793 | 0.637 | 0.733 | 0.800 |

Table 2: Global concept importance attribution comparison on CUB

| Model | Method | Concept Accuracy | Global (per-class) | | |
|---|---|---|---|---|---|
| | | | R@30 | R@40 | R@50 |
| Inception v3 | TCAV | 0.709 | 0.619 | 0.742 | 0.822 |
| | global-CG | 0.791 | 0.790 | 0.894 | 0.944 |
| Resnet50 | TCAV | 0.727 | 0.592 | 0.692 | 0.775 |
| | global-CG | 0.793 | 0.827 | 0.915 | 0.951 |
| Vgg16 bn | TCAV | 0.703 | 0.628 | 0.747 | 0.832 |
| | global-CG | 0.793 | 0.872 | 0.935 | 0.961 |

**Ablation study on model layers.** We performed an ablation study on performing attribution with different input representations, as given by different model layers. We observe that for every input representation in every model, CG consistently outperforms CAV in concept importance attribution, both locally and globally. For CG, the performance trend peaks in the penultimate layer representation. While finetuning more layers may lead to better concept prediction, it does not necessarily translate into better attribution. There is a trade-off between the concept model $g$ capturing the concept well (finetuning more layers better) and utilizing the input more similarly to the target model $f$ (finetuning less layers better). In this case, the input representation in the penultimate layer retains sufficient information to capture the concept labels. For CAV, the performance is generally better in deeper layers since concepts are more linearly separable in the latter layers. Unlike the non-linearity of CG, CAV is unable to exploit the more concept information contained in the representation of earlier layers.

**Ablation study on normalization schemes.** We compared different methods of calculating the concept gradients with various gradient normalization schemes. We ran all experiments on Inception v3 with finetuned layers `Mixed_7b+`. The results are presented in Table 3. Recall that CG is exactly inner product normalized with squared concept norm, IP represents pure inner product between $\nabla f(x)$ and $\nabla g(x)$ without normalization, GC with normalized $\nabla g(x)$, and cosine with both normalized $\nabla f(x)$ and $\nabla g(x)$. We observe that all methods performs equally well. This supports the argument that the gradient norms in trained neural network are well-behaved and it is less common to encounter cases where normalization influences the attribution results significantly.

Table 3: Comparison of CG with different normalization schemes

| Scheme | Formula | Local | | | Global | | |
|---|---|---|---|---|---|---|---|
| | | R@30 | R@40 | R@50 | R@30 | R@40 | R@50 |
| CG | $\frac{\nabla g(x)^T}{\|\nabla g(x)\|^2} \cdot \nabla f(x)$ | 0.612 | 0.718 | 0.790 | 0.783 | 0.907 | 0.949 |
| IP | $\nabla g(x)^T \cdot \nabla f(x)$ | 0.601 | 0.718 | 0.801 | 0.783 | 0.907 | 0.949 |
| GC | $\frac{\nabla g(x)^T}{\|\nabla g(x)\|} \cdot \nabla f(x)$ | 0.610 | 0.720 | 0.799 | 0.783 | 0.907 | 0.949 |
| Cosine | $\frac{\nabla g(x)^T}{\|\nabla g(x)\|} \cdot \frac{\nabla f(x)}{\|\nabla f(x)\|}$ | 0.610 | 0.720 | 0.799 | 0.783 | 0.907 | 0.949 |

## 4.2 QUALITATIVE ANALYSIS

The purpose of this experiment is to provide intuition and serve as a sanity check by visualizing instances and how CG works.

**Dataset.** We conducted the experiment on the Animals with Attributes 2 (AwA2) dataset (Xian et al., 2018), an image classification dataset with 37k animal images, 50 animal classes, and 85 binary attributes for each class. These concepts cover a wide range of semantics, from low-level colors and

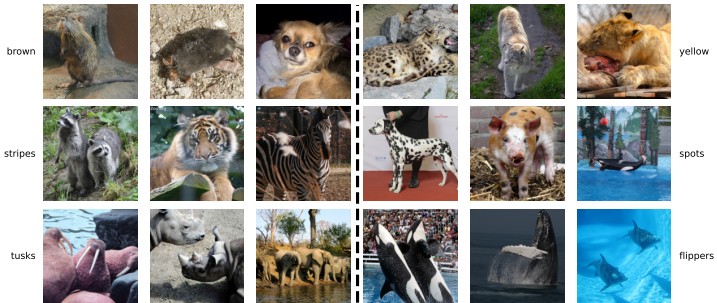

Figure 3: Visualization of instances with highest CG attributed importance (AwA2 validation set) for each concept (top 1 instance in the top 3 classes per concept). CG is capable of handling low level (colors), middle level (textures), and high level (body components) concepts simultaneously.

textures, to high-level abstract descriptions (e.g. "smart", "domestic"). We further filtered out 60 concepts that are visible in the input images to perform interpretation.

**Evaluation.** The evaluation is performed on the validation set. Fig 3 visualizes the instances with the highest CG importance attribution for 6 selected concepts, filtering out samples from the same class (top 1 instance in the top 3 classes). The concepts are selected to represent different levels of semantics. The top row contains colors (low-level), the middle row contains textures (medium-level), and the bottom row contains body components (high-level). Observe that CG is capable of handling different levels of semantics simultaneously well, owing to the expressiveness of non-linear concept model $g$. Additionally, we presented **randomly sampled** instances from the validation set and listed top-10 most important concepts as attributed by CG (see the appendix). We intentionally avoided curating the visualization samples to demonstrate the true importance attribution performance of CG. The most important concepts for each instance passed the sanity check. There are no contradictory concept-class pairings and importance is attributed to concepts existent in the images.

### 4.3 CASE STUDY ON MORTALITY RISK OF MYOCARDIAL INFARCTION COMPLICATIONS

The purpose of the case study is to demonstrate the effectiveness of CG in critical domain applications beyond classification tasks on natural-image data.

**Dataset.** We experimented on the Myocardial infarction complications database. The database consists of 1,700 entries of patient data with 112 fields of input tabular features, 11 complication fields for prediction, and 1 field of lethal outcome also for prediction. The input features are measurements taken when the patients were admitted to the hospital due to myocardial infarction as well as past medical records. The target models predict lethal outcome given the 112 input fields. The concept models predict the complications given the same 112 input fields.

**Evaluation.** Our goal is to interpret the lethal outcome with the complications and compare our interpretation with existing literature regarding how each complication affects the risk of death. We expect good interpretations to assign high relevance to complications that pose high mortality risk. Table 4 shows the global CG scores (aggregated by averaging local CG) as well as the excerpted description of the complication mortality risk in the existing medical literature. The severity of descriptions in the medical literature is largely aligned with the CG scores. The highest risk complications are attributed the most importance (e.g. relapse of MI) while the lower risk complications are attributed the least importance (e.g. post-infarction angina). This supports CG as an effective method for real-life practical use for interpreting models in critical domains. We also provided TCAV scores (aggregated by averaging local CAV) for comparison, which is largely aligned with CG with few exceptions (e.g. chronic heart failure) where the CAV interpretation deviate from literature. The full table can by found in Appendix J.3.

## 5 RELATED WORK

Our work belongs to post-hoc concept-based explanations. While training with self-interpretable models (Bouchacourt & Denoyer, 2019; Chen et al., 2019; Lee et al., 2019; Wang & Rudin, 2015)

Table 4: Mortality risk attribution with respect to a subset of myocardial infarction complications and comparison with existing medical literature

| Complication | CG | TCAV | Excerpted mortality risk description from medical literature |
|---|---|---|---|
| Relapse of MI | 3.47 | 2.55 | Recurrent infarction causes the most deaths following myocardial infarction with left ventricular dysfunction. (Orn et al., 2005) |
| Chronic heart failure | 3.27 | -1.26 | The mortality rate in a group of patients with class III and IV heart failure is about 40% per year, and half of the deaths are sudden. (Bigger, 1987) |
| Myocardial rupture | 1.62 | 6.52 | Myocardial rupture is a relatively rare and usually fatal complication of myocardial infarction (MI). (Shamshad et al., 2010) |
| Ventricular fibrillation | 0.91 | 1.90 | Patients developing VF in the setting of acute MI are at higher risk of in-hospital mortality. (Bougouin et al., 2014) |
| Dressler syndrome | 0.32 | -2.85 | The prognosis for patients with DS is typically considered to be quite good. (Leib et al., 2017) |
| Post-infarction angina | -1.40 | -2.85 | After adjustment, angina was only weakly associated with cardiovascular death, myocardial infarction, or stroke. (Eisen et al., 2016) |

is recommended if the use case allows, often times concepts are not known beforehand which makes the practicality of post-hoc explanations more widely adoptable. Other classes of post-hoc explanations include featured-based explanations (Zintgraf et al., 2017; Petsiuk et al., 2018; Dabkowski & Gal, 2017; Shrikumar et al., 2017; Sundararajan et al., 2017), counterfactual explanations(Dhurandhar et al., 2018; Hendricks et al., 2018; van der Waa et al., 2018; Goyal et al., 2019b; Joshi et al., 2019; Poyiadzi et al., 2020; Hsieh et al., 2021), and sample-based explanations (Bien & Tibshirani, 2011; Kim et al., 2016; Koh & Liang, 2017; Yeh et al., 2018; Pruthi et al., 2020; Khanna et al., 2018). Our work considers the gradient from prediction to concepts, which is in spirit connected to feature explanations which considers the gradient from prediction to feature inputs (Zeiler & Fergus, 2014; Ancona et al., 2017).

Concept-based explanations aims to provide human-centered explanations which answer the question "does this human understandable concept relates to the model prediction?". Some follows-up for concept-based explanations include when are concept sufficient to explain a model (Yeh et al., 2019), computing interventions on concepts for post-hoc models (Goyal et al., 2019a) and self-interpretable models (Koh et al., 2020), combining concept with other feature attributions (Schrouff et al., 2021), unsupervised discovery of concepts (Ghorbani et al., 2019; Yeh et al., 2020; Ghandeharioun et al., 2021), and debiasing concepts (Bahadori & Heckerman, 2020). The most similar work to ours is the work of Chen et al. (2020), which is motivated by the issue of CAV that concept does not necessarily lie in the linear subspace of some activation layer. They address this problem by training a self-interpretable model and limits the concepts to be whitened. On the contrary, our work address the non-linear concept problem of CAV in the post-hoc setting by learning a non-linear concept component and connects to the activation space via chain rule. Another work that considers nonlinear modeling of concepts is TCAR (Crabbé & van der Schaar, 2022). They modeled concepts with non-linear kernel functions and calculate relevance score via the concept function output magnitude. However, merely considering the function output ignores interactions between the target and concept function and potentially leads to explaining spurious correlations.

## 6 CONCLUSION

We revisited the fundamental assumptions of CAV, one of the most popular concept-based, gradient interpretation methods. We tackled the problem from a mathematical standpoint and proposed CG to directly evaluate the gradient of a given target model with respect to concepts. Our insight explains the success of CAV, which is a linear special case of CG. Empirical experiments demonstrated that CG outperforms CAV on real datasets and is useful in interpreting models for critical domain applications. Currently CG depends on the representation of input. Devising an input representation invariant method is an interesting future direction. CG also requires user-specified concepts to function. Integrating automatic novel concepts discovery mechanisms into the method may be useful in applications with insufficient domain knowledge.

## ACKNOWLEDGEMENT

This work is supported in part by NSF under IIS-2008173, IIS-2048280, an Okawa research grant, NIH NIGMS MIRA (1R35GM146735), UCLA California NanoSystems Institute and the Noble Family Innovation Fund. I would also like to thank Karen who provided immense emotional support and offered insightful suggestions.

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

# A    LIMITATIONS

CG is a gradient-based interpretation methods, which is can only be applied to differentiable white-box models. Gradient-based methods convey how small changes in the input affect the output via gradients. Larger changes in the input requires intervention-based causal analysis to predict how the output is affected. As a modification upon CAV, CG also requires users to specify concepts with sufficient representative data samples or close-form concept functions. Sufficient amount of data is more important for CG to prevent the nonlinear function to overfit. Automatic discovery of novel concepts requires introducing other tools. Finally, CG requires fitting non-linear concept models if the concept is provided in the form of representative data samples. This might be computationally intensive for complex non-linear models (e.g. neural networks). The quality of interpretation highly depends on how accurately the concept is captured by the concept model.

# B    GENERAL CG FOR MULTIPLE CONCEPTS

Here we derive CG in the general case when the concept function maps $x \in \mathbb{R}^d$ to $m$ concepts $g : \mathbb{R}^d \to \mathbb{R}^m$. Recall the definition of CG:

$$R_{\mathrm{CG}}(x; f, g) = \nabla g(x)^{\dagger} \nabla f(x)$$

Note that when computing the gradient of a multivariate function such as $f$, we follow the convention that $\nabla f(x) \in \mathbb{R}^{d \times k}$ where the $(\nabla f(x))_{ij} = \frac{\partial f_j(x)}{\partial x_i}$. And $\nabla g(x)^{\dagger} \in \mathbb{R}^{m \times d}$ is the pseudo-inverse of $\nabla g(x)$. $R_{\mathrm{CG}}(x)$ will thus be an $m \times k$ matrix and its $(i, j)$ element measures the contribution of concept $i$ to label $j$.

Let us first consider the scenario where $\nabla g(x)$ is invertible. In this case, there exists an unique function $g^{-1}(c)$ mapping $c$ to $x$ locally around $\hat{c}$. By chain rule CG$(\hat{x})$ is equivalent to the derivative of $y$ with respect to $c$:

$$\left.\frac{\partial y}{\partial c}\right|_{c=\hat{c}} = \left.\frac{\partial f(g^{-1}(c))}{\partial c}\right|_{c=\hat{c}} = \nabla g^{-1}(\hat{c}) \nabla f(g^{-1}(\hat{c})) = (\nabla g(\hat{c}))^{-1} \nabla f(\hat{x}) = \mathrm{CG}(\hat{x}).$$

Now let us consider the scenario when $\nabla g(x)$ is not invertible. For simplicity, we assume $\nabla g(x)$ has full column rank, which implies $g$ is surjective (this is always true when $m = 1$). Otherwise we can directly constrain $c$ within the row space of $\nabla g(x)$ and the arguments remain valid.

Our goal is to construct a mapping from $c$ to $x$ and analyze the gradient. However, there are infinitely many functions from $c$ to $x$ that can locally inverse $g(x)$ since the dimension of concept ($m$) could be much smaller than input dimension $d$. Despite an infinite number of choices, we show the gradient of such function always follows a particular form:

**Theorem 1.** *Consider a particular point $\hat{x}$ with $\hat{c} = g(\hat{x})$. Let $h : \mathbb{R}^m \to \mathbb{R}^d$ be a smooth and differentiable function mapping $c$ to $x$ and satisfy $g(h(c)) = c$ locally within the $\epsilon$-ball around $\hat{c}$, then the gradient of $h$ will take the form of*

$$\nabla h(\hat{c}) = \nabla g(\hat{x})^{\dagger} + G_{\perp}, \tag{7}$$

*where any row vector of $G_{\perp}$ belongs to null($\nabla g(x_0)^T$) (null space of $\nabla g(x_0)^T$).*

The proof of the theorem is deferred to the appendix. Intuitively, this implies the gradient of $h$ will take a particular form in the space of $\nabla g(x)^T$ while being arbitrary in its null space since any change in the null space cannot locally affect $c$. We can verify that $g(x)$ is locally unchanged in the null space of $\nabla g(x)^T$, as

$$g(x + G_{\perp}) \approx g(x) + \nabla g(x)^T G_{\perp} = g(x).$$

This is saying there are multiple choices in $\Delta x$ to achieve the same effect on $c$, since any additional perturbation in null($\nabla g(x)^T$) won't make any change to $c$ locally. For example, when we want to change the concept of "color" in an image by $\Delta x$, we can have $\Delta x$ only including the minimal change (e.g., only changing the color), or have $\Delta x$ including change of color and any arbitrary change to another orthogonal factor (e.g., shape). For concept-based attribution, it is natural to consider the minimal space of $x$ that can cover $c$, which corresponds to setting $G_{\perp}$ as 0 in (7).

If we pick such $h$ then $\nabla h(\hat{c}) = \nabla g(\hat{x})^\dagger$, so we can represent $y$ as a function of $c$ locally by $y = p(c) := f(h(c))$ near $\hat{c}$. By chain rule we then have

$$\nabla p(\hat{c}) = \nabla h(\hat{c}) \nabla f(\hat{x}) = \nabla g(\hat{x})^\dagger \nabla f(\hat{x}) = R_{\text{CG}}(\hat{x}).$$

In summary, although there are infinitely many functions from $c$ to $y$ since the input space has a larger dimension than the concept space, if we consider a subspace in $x$ such that change of $x$ will affect $c$ (locally) and ignore the orthogonal space that is irrelevant to $c$, then any function mapping from $c$ to $y$ through this space will have gradient equal to Concept Gradients defined in (3).

### B.1  SHOULD WE EXPLAIN MULTIPLE CONCEPTS JOINTLY OR INDIVIDUALLY?

When attributing the prediction to multiple concepts, our flexible framework enables two options: 1) treating each concept independently using single-concept attribution formulation (**??**) 2) Combining all the concepts together into $c \in \mathbb{R}^m$ and run CG. Intuitively, option 2 takes the correlations between concepts into account while option 1 does not. When both concepts A and B are important for prediction but concept A is slightly more important than concept B, option 1 will identify both of them to be important while option 2 may attribute the prediction to concept B only. For example, when $y = x_0 + x_1, c_0 = x_0, c_1 = x_0 + 0.1x_1$, option 1 will identify both concepts to be important, while option 2 will produce negative score for $c_0$. We further compared these two options on real datasets. The results are presented in Table 5. We verified that empirically applying pseudo-inverse individually (independently) for each concept is better aligned with the attributes labeled by humans.

Table 5: Comparison of different CG pseudo-inverse calculations

| Method | Local | | | Global | | |
|---|---|---|---|---|---|---|
| | R@30 | R@40 | R@50 | R@30 | R@40 | R@50 |
| Joint | 0.345 | 0.440 | 0.530 | 0.409 | 0.511 | 0.607 |
| Independent | 0.612 | 0.718 | 0.790 | 0.783 | 0.907 | 0.949 |

## C  EXAMPLE FOR DEMONSTRATING DIFFERENT CG CALCULATION METHODS

We use a simple toy example to explain the difference between option 1 and 2. Assume $y = x_0 + x_1$ and two concepts $c_0 = x_0, c_1 = x_0 + 0.1x_1$. Since these relationships are all linear, concept gradients will be invariant to reference points. The results computed by option 1 and 2 are

Option 1 (individually apply (**??**)): $R_{c_0,y} = 1, R_{c_1,y} \approx 1.01$

Option 2 (apply CG jointly by (3)): $R_{c_0,y} = -9, R_{c_1,y} = 10$

At the first glance, it might be counterintuitive as to why the contribution of $c_0$ is negative to $y$ with option 2. But in fact there exists a unique function $y = -9c_0 + 10c_1$ mapping $c$ to $y$ (when jointly considering two concepts), which leads to negative contribution of $c_0$. This shows that when considering two concepts together as a joint function, the gradient is trying to capture the effect of one concept with respect to others, which may be non-intuitive to human.

## D  EXAMPLE FOR SELECTING PSEUDO-INVERSE INSTEAD OF OTHER INVERSE MATRICES

For the calculation of CG, multiple inverse matrices exist when $\nabla g(x)$ is not invertible. We gave a high level example in Section B of why it makes sense to apply the pseudo-inverse. In this example, we provide a concrete example with numbers to further elaborate.

Suppose the raw input x has two dimensions: $x_1$ indicates the feature "color" and $x_2$ indicates the feature "shape". There's only one single concept $c$ corresponds to "color", so $c = g(x) = x_1$. The target model is $y = f(x) = x_1 - x_2$.

Let's consider a particular input $\hat{x} = [1, 0]$ and $\hat{c} = 1$. Clearly a small perturbation to "color" can lead to linear change to output with slope 1, so the concept attribution score should be 1. Now let's follow the derivation of CG in this example to show why we choose the minimum-norm solution.

Centered at $\hat{c}$, there can be infinite number of (linear) mappings from $c$ to $x$, but all of them will follow the form according to Theorem 1:

$$h(c) = h(\hat{c}) + \nabla h(\hat{c})(c - \hat{c})$$
$$\nabla h(\hat{c}) = \nabla g(\hat{x})^\dagger + G_\perp$$

In our example, $h$ can be written explicitly as follows:

$$h(c) = [1, 0] + ([1, 0] + [0, \alpha])(c - 1)$$
$$= [c, \alpha(c - 1)], \quad \forall \alpha$$

This corresponds to Theorem 1, where $\nabla g(\hat{x})^+ = [1, 0]$ and $G_\perp$ is the set of $[0, \alpha]$ with any $\alpha$. These functions all set $x_1 = c$ but will set $x_2 = \alpha(c - 1)$ with arbitrary $\alpha$. Conceptually, this is saying there are infinite number of ways to form the mapping from "color" ($c$) to input ($x$): we always change "color" according to $c$ but we can arbitrarily change the other orthogonal feature "shape" ($x_2$) according to color. Concept Gradients is trying to see how the change of $c$ affect output $y$ while **keeping all the other orthogonal factors fixed**. Therefore, we want to find the mapping that has minimal perturbation to $x$ when changing $c$, corresponding to keeping all other orthogonal factors fixed. This is the reason to choose the minimal norm solution in pseudo inverse.

In this example, our method sets $\alpha = 0$ and will lead to $h(c) = [1, 0]$. If $\alpha$ is set to some non-zero values in the pseudo-inverse, it will lead to the following score:

$$\frac{\partial f(h(\hat{c}))}{\partial c} = \langle [1, \alpha], [1, -1] \rangle = 1 - \alpha$$

which will give the wrong attribution score for any $\alpha \neq 0$.

## E  PROOF

**Theorem 2.** *Let $h : \mathbb{R}^m \to \mathbb{R}^d$ be a smooth and differentiable function mapping $c$ to $x$ and satisfy $g(h(c)) = c$ locally within an $\epsilon$-ball around $c_0$. Then the gradient of $h$ will take the form of*

$$\nabla h(c_0) = \nabla g(x_0)^\dagger + g_\perp, \tag{8}$$

*where $g_\perp$ is in the null space of $\nabla g(x_0)^T$.*

*Proof.* Assume $h : \mathbb{R}^m \to \mathbb{R}^d$ is a function mapping $c$ to $x$. By Taylor expansion we have

$$g(x') = g(x) + \nabla g(x)^T (x' - x) + O(\|x' - x\|^2) \tag{9}$$
$$h(c') = h(c) + \nabla h(c)^T (c' - c) + O(\|c' - c\|^2), \tag{10}$$

where $x' = h(c')$ and $x = h(c)$. Therefore

$$g(x') = g(x) + \nabla g(x)^T (h(c') - h(c)) + O(\|h(c') - h(c)\|^2)$$
$$= g(x) + \nabla g(x)^T \nabla h(c)^T (c' - c) + O(\|\nabla g(x)\|^2 \|c' - c\|^2) + O(\|h(c') - h(c)\|^2).$$

The last two terms are both $O(\|c - c'\|^2)$ since $\nabla g(x)$ is a constant and $h$ is smooth (thus locally Lipschitz), so

$$c' - c = \nabla g(x)^T \nabla h(c)^T (c' - c) + O(\|c - c\|^2). \tag{11}$$

Therefore, $\nabla g(x)^T \nabla h(c)^T = I$, which means $\nabla h(c) = \nabla g(x)^\dagger + g_\perp$. $\qquad\square$

### E.1  DEFINITION OF CONCEPT GRADIENTS

Given input space $\mathcal{X}$ and two differentiable functions $f : \mathcal{X} \to \mathcal{Y}$ and $g : \mathcal{X} \to \mathcal{C}$, where $\mathcal{Y}$ is the target label space and $\mathcal{C}$ is the concept label space. We define the concept gradients of $x \in \mathcal{X}$ to attribute the prediction of the model to the concepts:

$$R_{\text{CG}}(x) := \nabla f(x) \cdot \nabla g(x)^\dagger, \tag{12}$$

where $\nabla g(x)^\dagger$ denotes the pseudo-inverse of $\nabla g(x)$. Let $y = f(x) \in \mathcal{Y}$ and $c = g(x) \in \mathcal{C}$. Essentially concept gradients approximate gradient-based saliency of $y$ to $c$, $h'(c)$, via chain rule. For the case where $\mathcal{X}$, $\mathcal{Y}$, and $\mathcal{C}$ are all scalar fields, the concept gradients exactly recover $h'(c)$ if $g'(x) \neq 0$ for all $x \in \mathcal{X}$,

Given input space $\mathcal{X}$ and two differentiable functions $f : \mathcal{X} \to \mathcal{Y}$ and $g : \mathcal{X} \to \mathcal{C}$, where $\mathcal{Y}$ is the target label space and $\mathcal{C}$ is the concept label space. Suppose there exist an unknown differentiable function $h : \mathcal{C} \to \mathcal{Y}$ s.t. $f = h \circ g$. Let $f'$, $g'$, and $h'$ denote the first-order derivatives. We define the concept gradients of $x \in \mathcal{X}$ as

$$R_{\mathrm{CG}}(x) := \nabla f(x) \cdot \nabla g(x)^\dagger$$

where $\nabla g(x)^\dagger$ denotes the pseudo-inverse of $\nabla g(x)$. Let $y = f(x) \in \mathcal{Y}$ and $c = g(x) \in \mathcal{C}$. Essentially concept gradients approximate gradient-based saliency of $y$ to $c$, $h'(c)$, via chain rule. For the case where $\mathcal{X}$, $\mathcal{Y}$, and $\mathcal{C}$ are all scalar fields, the concept gradients exactly recover $h'(c)$ if $g'(x) \neq 0$ for all $x \in \mathcal{X}$,

$$h'(c) = \frac{\mathrm{d}y}{\mathrm{d}c} = \frac{\mathrm{d}y}{\mathrm{d}x} \cdot \frac{\mathrm{d}x}{\mathrm{d}c} = \frac{\mathrm{d}y}{\mathrm{d}x} \cdot (\frac{\mathrm{d}c}{\mathrm{d}x})^{-1} = f'(x) \cdot \frac{1}{g'(x)} = R_{\mathrm{CG}}(x)$$

We now generalize $\mathcal{X}$ to a n-dimensional vector space. Since $f$, $g$, and $h$ are differentiable, we can perform Taylor expansion around $x$ and $c$

$$f(x') = f(x) + \nabla f(x)(x' - x) + O((x' - x)^2) \tag{13}$$
$$g(x') = g(x) + \nabla g(x)(x' - x) + O((x' - x)^2) \tag{14}$$
$$h(c') = h(c) + h'(c)(c' - c) + O((c' - c)^2) \tag{15}$$

Let us denote $\Delta x = x' - x$. We can plug Eq 14 into Eq 15

$$
\begin{aligned}
h(g(x')) - h(c) &= h'(c)(g(x') - c) + O((g(x') - c)^2) \\
&= h'(c) \cdot \left( g(x) + \nabla g(x)\Delta x + O(\Delta x^2) - c \right) + O((g(x') - c)^2) \\
&= h'(c) \cdot \left( \nabla g(x)\Delta x + O(\Delta x^2) \right) + O((g(x') - c)^2) \\
&\approx h'(c) \cdot \left( \nabla g(x)\Delta x \right)
\end{aligned}
$$

$$
\begin{aligned}
f(x') - f(x) &= \nabla f(x)\Delta x + O(\Delta x^2) \\
&\approx \nabla f(x)\Delta x
\end{aligned}
$$

$$h(g(x')) - h(c) = f(x') - f(x)$$
$$h'(c) \cdot \left( \nabla g(x)\Delta x \right) \approx \nabla f(x)\Delta x$$

Let us denote the set of right inverses for $\nabla g(x)$ as $G_r^{-1}(x)$.

$$G_r^{-1}(x) = \{\nabla g(x)^\dagger + g_\perp^T, \forall g_\perp : \langle \nabla g(x), g_\perp \rangle = 0\}$$

By definition for all $g_r^{-1} \in G_r^{-1}(x)$,

$$h'(c) \cdot \Delta x \approx \nabla f(x) \cdot g_r^{-1} \cdot \Delta x$$

If $\nabla g(x)$ is invertible, $G_r^{-1}(x) = \{\nabla g(x)^\dagger\}$ and the equality exactly holds

$$h'(c) = \nabla f(x) \cdot \nabla g(x)^\dagger = \nabla f(x) \cdot \nabla g(x)^{-1}$$

If $\nabla g(x)$ is not invertible, $g_r^{-1}$ is not unique and infinitely many right inverses exist for $\nabla g(x)$. In this case, how does the selection of right inverse relate to interpretation? For interpretation, the goal is to attribute a small change $\Delta c$ via $g_r^{-1}$ to a small change $\Delta x$. Non-invertibility of $\nabla g(x)$ implies there are many ways we could perform the attribution. Consider the alignment between some $g_r^{-1} = \nabla g(x)^\dagger + g_\perp^T$ and $\nabla g(x)$,

$$\frac{\nabla g(x) \cdot (\nabla g(x)^\dagger + g_\perp^T)}{\|\nabla g(x)\| \cdot \|\nabla g(x)^\dagger + g_\perp^T\|} = \frac{\nabla g(x) \cdot \nabla g(x)^\dagger}{\|\nabla g(x)\| \cdot \|\nabla g(x)^\dagger + g_\perp^T\|} \leq \frac{\nabla g(x) \cdot \nabla g(x)^\dagger}{\|\nabla g(x)\| \cdot \|\nabla g(x)^\dagger\|}$$

We argue that the best interpretation is when the attribution is faithful to the relation $g$ between $\mathcal{X}$ and $\mathcal{C}$, i.e., when $g_r^{-1}$ is best aligned with $\nabla g(x)$. Observe that $\nabla g(x)^\dagger$ is in the same direction as $\nabla g(x)$. On the other hand, any right inverse with $g_\perp \neq 0$ is attributing *some* proportion of d$c$ to d$x$ in a direction that is orthogonal to $\nabla g(x)$. Thus, the best choice for $g_r^{-1}$ is $\nabla g(x)^\dagger$.

### E.2 Connection with CAV

CAV is a special case of concept gradients with $g$ restricted to linear functions. Let $\boldsymbol{v}_c$ denote the concept activation vector associated with concept $c$. CAV defines the conceptual sensitivity $S$ as the inner product of the input gradient and concept activation vector,

$$R_{\mathrm{CAV}}(x) := \nabla f(x) \cdot \frac{\boldsymbol{v}_c}{\|\boldsymbol{v}_c\|}$$

If $g$ is restricted to linear functions,

$$g(x) = \boldsymbol{v}_c^T \cdot x + b_c$$

for some constant bias $b_c$. Concept gradients is equivalent to CAV conceptual sensitivity normalized by the norm of the concept activation vector,

$$R_{\mathrm{CG}}(x) = \nabla f(x) \cdot \nabla g(x)^\dagger = \nabla f(x) \cdot (\boldsymbol{v}_c^T)^\dagger = \nabla f(x) \cdot \frac{\boldsymbol{v}_c}{\|\boldsymbol{v}_c\|^2} = \frac{R_{\mathrm{CAV}}(x)}{\|\boldsymbol{v}_c\|}$$

Thus, if the concept can be accurately modeled by a linear function, CAV is capable of retrieving the concept gradients. However, in general the linear separability assumption does not hold. In contrast, concept gradients consider general function classes for $g$, which better captures the relation between $\mathcal{X}$ and $\mathcal{C}$. Given accurately modeling the concept with $g$ is a necessary condition for correct interpretation, concept gradients is superior to CAV.

## F Alternative perspective for layer selection

We can also view layer selection in a typical bias-variance tradeoff perspective. If we selected a later layer to evaluate CG, we are biased towards using a representation of $x$ that is optimized for predicting the target $y$, not $c$. However, since the information consists in the representation is less, we also enjoy the benefit of less variance. On the other hand, if we selected an earlier layer to evaluate CG, then we suffer less from the bias (towards $y$) but is penalized with higher variance due to abundance of information. The optimal layer is where the representation of $x$ is capable of predicting the concept (minimized bias) while no redundant information is available (minimized variance).

We verified the bias-variance tradeoff hypothesis with experiments. More bias (with respect to target labels) in later layers is confirmed with the observation that finetuning more layers yields higher concept prediction accuracy (see Fig 4). Less variance in later layers is confirmed with the experiment below. We repeated the CUB experiments on the Inception v3 model with 5 different random seeds and evaluated the variance of the gradient $\nabla g(x)$ over repeated trials, averaged over all data points. Specifically, the gradients for models finetuned starting from different layers are evaluated on the same layer (`Mixed_6d`) for fair comparison. The results are shown in Fig 5 and confirmed the variance hypothesis.

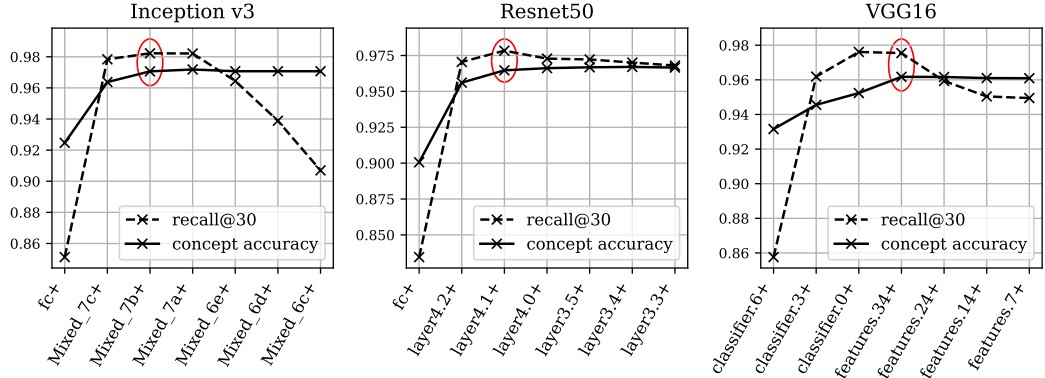

Figure 4: Concept prediction accuracy and concept attribution recall when finetuning starting from different layers of the model. Finetuning more layers leads to higher concept prediction accuracy.

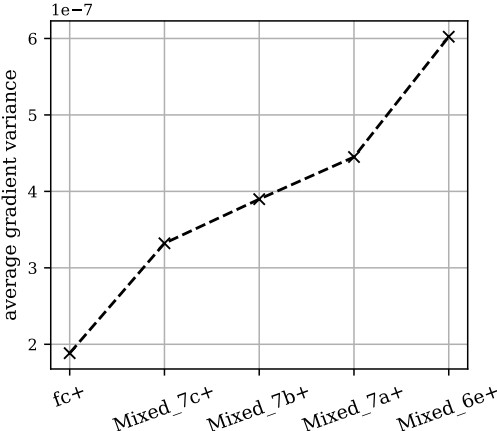

Figure 5: Variance of gradients finetuning starting from different layers. The variance is higher when finetuning starting from earlier layers.

## G ABLATION STUDY ON CONCEPT MODEL CONFIGURATION

Experiments in Section 4.1 and 4.2 shares the same model architecture for the target and concept model. The two design choices here are 1) using the same model architecture and 2) warm-starting the training of concept models with target model weights. The choices are rather straightforward since both models need to share the same input feature representation for the gradients with respect to the input layer to be meaningful. Nevertheless, we conducted an ablation study on the CUB experiment to verify how using different model architectures different weight initialization for the concept model affects interpretation. In this case, CG needs to be evaluated in the input layer, the only layer where the feature representation is shared between target and concept models.

The model architecture alabltion study results are presented in Table 6. The CG scores are all evaluated in the input layer. Evidently, using the same model architecture for both the target and concept models is crucial for good interpretation quality with CG. The degradation of interpretation when using different architectures may be caused by mismatched usage of input feature representation between models. The concept model weight initialization results are presented in Table 7. Using target model weights as initialization outperforms using ImageNet pretrained weights significantly. The more similar the pretrained task is to the downstream task is, the better the finetuned performance. In this case, the concept prediction accuracy suggests the target model task of bird classification is

a better pretraining task for predicting bird concepts, allowing the concepts to be better captured by the concept function (higher accuracy). This naturally leads to better interpretation results.

Table 6: Ablation study on concept model architecture

| Target model | Concept model | R@30 | R@40 | R@50. |
|---|---|---|---|---|
| Inception v3 | Inception v3 | 0.872 | 0.931 | 0.957 |
| Inception v3 | Resnet50 | 0.535 | 0.620 | 0.684 |
| Inception v3 | VGG16 | 0.486 | 0.561 | 0.619 |
| Resnet50 | Resnet50 | 0.933 | 0.976 | 0.988 |
| Resnet50 | Inception v3 | 0.623 | 0.705 | 0.757 |
| Resnet50 | VGG16 | 0.554 | 0.631 | 0.692 |
| VGG16 | VGG16 | 0.955 | 0.986 | 0.994 |
| VGG16 | Inception v3 | 0.525 | 0.606 | 0.664 |
| VGG16 | Resnet50 | 0.609 | 0.698 | 0.758 |

Table 7: Ablation study on concept model weight initialization

| Weight initialization | Concept accuracy | R@30 | R@40 | R@50. |
|---|---|---|---|---|
| ImageNet pretrained | 0.916 | 0.577 | 0.670 | 0.739 |
| target model pretrained | 0.972 | 0.872 | 0.931 | 0.957 |

## H ABLATION STUDY ON THE EXPRESSIVENESS OF CONCEPT MODEL CLASS

In most of this study, the concept model is implemented by finetuning the target model to predict the concepts instead. When using an input representation in deeper layers, the portion of finetuned model would be relatively less, yielding a simpler concept model. Our hypothesis is input representation in deeper layers generally contains higher level semantics and thus only requires simpler models to capture the concept.

To test this hypothesis, we explored whether using a more expressive function class for the concept model benefits the attribution result. We focused on the ResNet architecture and constructed concept models of different complexities. We experimented on input representations in deeper layers (`layer4.1` and `layer4.2`) where the original concept model is simpler and may not be expressiveness enough to capture the concepts. To increase the complexity of a ResNet model, we duplicated ResNet residual blocks. The more times a block is duplicated, the deeper and more expressive a model is.

Table 8 shows the results. The number of duplication represents how many time the selected residual blocks is duplicated for the concept model. As increasing the complexity of the model does not lead to better concept prediction accuracy, the attribution performance (recall) is not improved. This confirmed our hypothesis for using a simpler (more complex) models for deeper (shallower) layer representations.

Table 8: Comparing concept models of different expressiveness. Increasing model expressiveness does not lead to increase in accuracy and thus does not translate to better attribution performance.

| Residual Block | Number of Duplication | Concept Accuracy | Local (per-sample) | | | Global (per-class) | | |
|---|---|---|---|---|---|---|---|---|
| | | | R@30 | R@40 | R@50 | R@30 | R@40 | R@50 |
| layer4.1 | 1 | 0.794 | 0.614 | 0.718 | 0.786 | 0.808 | 0.909 | 0.948 |
| | 2 | 0.795 | 0.605 | 0.705 | 0.775 | 0.810 | 0.901 | 0.944 |
| | 3 | 0.791 | 0.597 | 0.697 | 0.767 | 0.811 | 0.904 | 0.944 |
| | 4 | 0.791 | 0.594 | 0.693 | 0.762 | 0.820 | 0.906 | 0.943 |
| layer4.2 | 1 | 0.793 | 0.627 | 0.726 | 0.794 | 0.827 | 0.915 | 0.951 |
| | 2 | 0.793 | 0.622 | 0.720 | 0.788 | 0.838 | 0.919 | 0.955 |
| | 3 | 0.794 | 0.613 | 0.710 | 0.777 | 0.838 | 0.918 | 0.952 |
| | 4 | 0.793 | 0.613 | 0.712 | 0.781 | 0.857 | 0.931 | 0.964 |

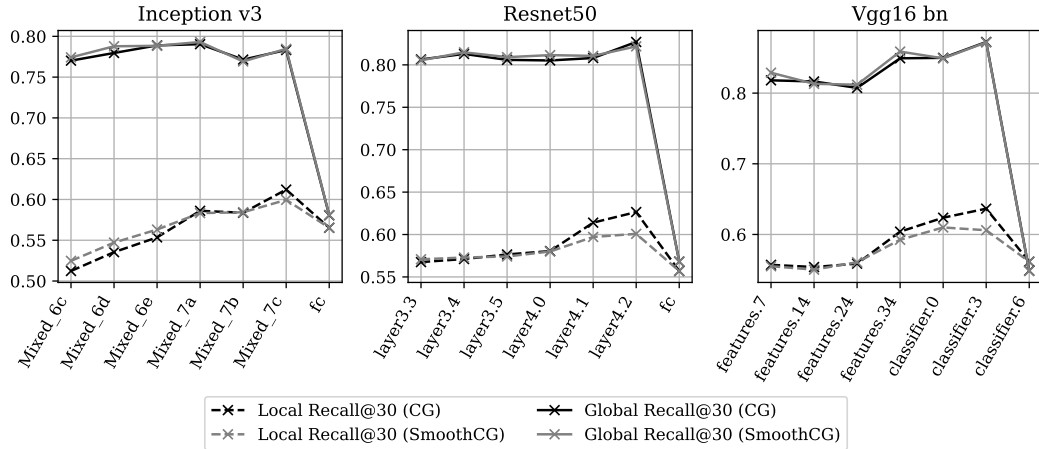

Figure 6: Comparing CUB concept recalls between CG and SmoothCG (stdev=0.01, n=8) for different input representations in various layers and architectures (left to right, deep to shallow layers). SmoothCG performs similarly to CG with marginal improvement in shallow layers.

## I  TO SMOOTH OR NOT TO SMOOTH?

It has been shown that relying on gradient saliency maps may yield misleading interpretation due to the high non-linearity in deep neural networks. There are many works on improving gradient-based interpretations with respect to the input features (Smilkov et al., 2017; Sundararajan et al., 2017; Shrikumar et al., 2017). Since CG also relies on input gradients for calculation, it is important to examine whether CG suffers from the same issues. In this ablation study, we compared SmoothCG, a variant of CG, that remedies inaccurate gradients.

Inspired by SmoothGrad (Smilkov et al., 2017), we proposed SmoothCG, where the attribution is averaged over the neighborhood $D$ near a data sample.

$$\text{SmoothCG}(x) := \mathbb{E}_{\epsilon \sim D} \text{CG}(x + \epsilon)$$

Smoothing over the neighborhood mitigates inaccurate interpretation caused by abrupt changes in input gradients in highly non-linear functions. It is also shown that randomized smoothing improves the robustness (Cohen et al., 2019). In practice, we take finite samples near an input instance with a predetermined perturbation distribution (e.g. Gaussian) and average their CG scores.

Figure 6 shows the concept recall comparison between CG and SmoothCG. SmoothCG is calculated by smoothing over 8 samples per instance, adding perturbations sampled from $\mathcal{N}(0, 0.01)$. In general the performances are similar. CG performs marginally better in deeper layers while SmoothCG in shallower ones, likely because gradients propagating to deeper layers are less accurate and thus benefit from the smoothing. However, the difference in performance is not consistent nor significant enough to justify the addition computation cost of SmoothCG.

## J  EXPERIMENT DETAILS

### J.1  ANIMAL WITH ATTRIBUTES 2 (AWA2)

**Data preprocessing** Since the original task is proposed for zero-shot classification, the class labels in the default training and validation set is disjoint. To construct a typical classification task, we combined all data together then performed a 80:20 split for the new training and validation set. During training, the input images are augmented by random color jittering, horizontal flipping, and resizing, then cropped to the default input resolution of the model architecture (299 for Inception v3, 224 for others). During evaluation, the input images are resized and center cropped to the input resolution.

The attribute labels provided in the dataset contains both concrete and abstract concepts. Some abstract concepts cannot be identified merely by looking at the input image (e.g. new world vs

old world). We filtered out 25 attributes that are not identifiable via the input image and used the remaining 60 attributes for interpretation.

**Training** We trained the target model $f$ with Inception v3 architecture with ImageNet pretrained weights (excluding the final fully connected layer). We optimized with Adam (learning rate of 0.001, beta1 of 0.9, and beta2 of 0.999) with weight decay 0.0004 and schedule the learning rate decay by 0.1 every 15 epochs until the learning rate reaches 0.00001. We trained the model for a maximum of 200 epochs and early stopped if the validation accuracy had not improved for 50 consecutive epochs. The validation accuracy of the trained target model is 0.947.

We trained the concept model $g$ by finetuning different parts of $f$ (freezing different layer model weights). We reweighted the loss for positive and negative class to balance class proportions. We optimized with Adam (learning rate of 0.01, beta1 of 0.9, and beta2 of 0.999) with weight decay 0.00004 and schedule the learning rate decay by 0.1 every 25 epochs until the learning rate reaches 0.00001. We trained the model for a maximum of 200 epochs and early stopped if the validation accuracy had not improved for 50 consecutive epochs.

**Evaluation** Visualization is conducted on the validation set. Finetuning from `Mixed_7a` is the latest layer that still predicted the concepts well. According to the layer selection guideline, we selected `Mixed_7a` to evaluate CG. We computed CG with the individual inverse method (**??**). Fig 7 shows random samples from the validation set and their top 10 rated concepts. These samples are intentionally randomly sampled as opposed to intentionally curated to provide an intuition of the true effectiveness of CG. In general, the retrieved highest ranked concepts are relevant with the input image. In terms of sanity check, there are no contradictions in the concepts (e.g. furry is never assigned to an whales).

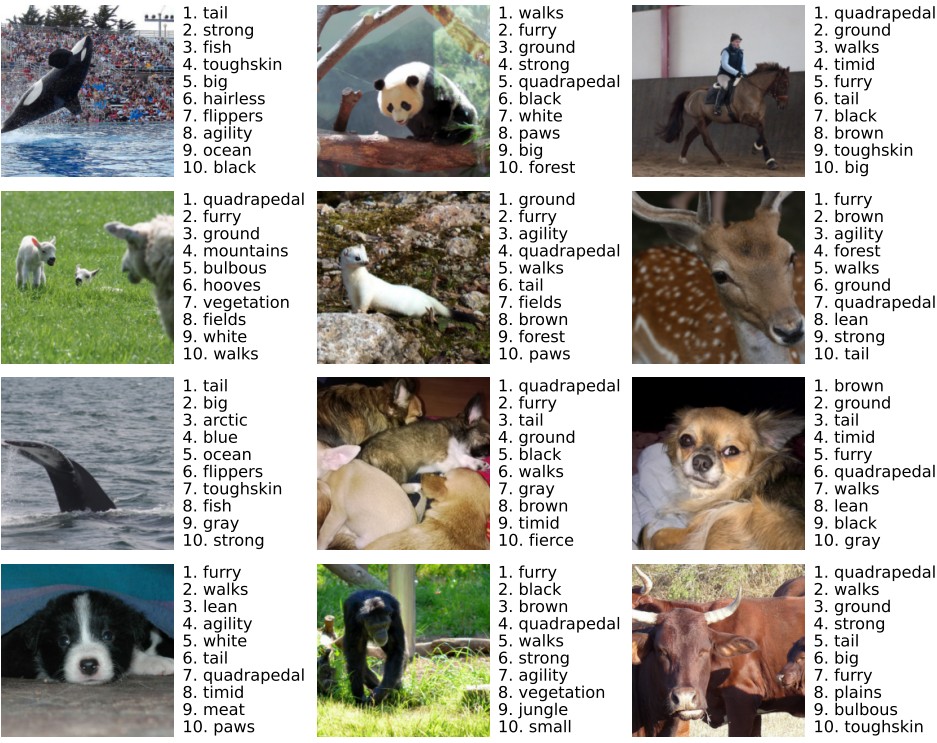

Figure 7: Visualization of randomly sampled instances (AwA2 validation set) and the most important concepts associated with their respective CG attribution (top 10 concept, 60 in total).

## J.2 CALTECH-UCSD BIRDS-200-2011 (CUB)

**Data preprocessing** Similar to the AWA2 experiments, the input images are augmented by random color jittering, horizontal flipping, and resizing, then cropped to the default input resolution of the

Table 9: Finetune layers for architectures

| Architecture | Finetuned layers |
|---|---|
| Inception v3 | `fc+, Mixed_7c+, Mixed_7b+, Mixed_7a+, Mixed_6e+, Mixed_6d+, Mixed_6c+` |
| Resnet50 | `fc+, layer4.2+, layer4.1+, layer4.0+, layer3.5+, layer3.4+, layer3.3+` |
| VGG16 | `classifier.6+, classifier.3+, classifier.0+, features.34+, features.24+, features.14+, features.7+` |

model architecture (299 for Inception v3, 224 for others) during training. During evaluation, the input images are resized and center cropped to the input resolution. We followed (Koh et al., 2020) procedure of removing attributes with insufficient data samples. A total of 112 attributes remains for conducting interpretation. The class attribute labels are assigned as instance labels, i.e., instances from the same class share the same attribute labels.

**Training** We trained the target model $f$ with three different CNN architectures: Inception v3, Resnet50, and VGG16 (with batch normalization), each with ImageNet pretrained weights (excluding the final fully connected layer). We searched for hyperparameters over a range of learning rates $(0.01, 0.001)$, learning rate schedules (decaying by 0.1 for every $15, 20, 25$ epochs until reaching $0.0001$), and weight decay $(0.0004, 0.00004)$. We optimized with the SGD optimizer. We trained the model for a maximum of 200 epochs and early stopped if the validation accuracy had not improved for 50 consecutive epochs. The validation accuracy of the trained target model is 0.797, 0.764, and 0.782 for the three models, respectively.

We trained the concept model $g$ by finetuning different parts of $f$ (freezing different layer model weights). The different layers we started to finetuned from for each model architecture is listed in Table 9. The plus sign in the table represents all layers after the specified layer are all finetuned while all layers prior to the specified layer have their weights kept frozen. We reweighted the loss for positive and negative class to balance class proportions. We searched for the hyperparameters and trained the model same as the target model $f$.

**Evaluation** Evaluation is conducted on the testing set. CG and CAV are evaluated in the layer prior to finetuning. We evaluated the recalls for $k = 30, 40, 50$ and generally the recall trend is consistent for all $k$s. These thresholds are chosen since the number of concepts with positive labels for each instance is in the range of 30 to 40.

## J.3 MYOCARDIAL INFARCTION COMPLICATIONS

Table 10: Mortality risk attribution with respect to myocardial infarction complications and comparison with existing medical literature

| Complication | CG | TCAV | Excerpted mortality risk description from medical literature |
|---|---|---|---|
| Relapse of MI | 3.47 | 2.55 | Recurrent infarction causes the most deaths following myocardial infarction with left ventricular dysfunction. (Orn et al., 2005) |
| Chronic heart failure | 3.27 | -1.26 | The mortality rate in a group of patients with class III and IV heart failure is about 40% per year, and half of the deaths are sudden. (Bigger, 1987) |
| Atrial fibrillation | 2.29 | 1.11 | AF increases the risk of death by 1.5-fold in men and 1.9-fold in women. (Benjamin et al., 1998) |
| Myocardial rupture | 1.62 | 6.52 | Myocardial rupture is a relatively rare and usually fatal complication of myocardial infarction (MI). (Shamshad et al., 2010) |
| Pulmonary edema | 1.51 | 2.21 | Pulmonary oedema in patients with acute MI hospitalized in coronary care units was reported to be associated with a high mortality of 38–57%. (Roguin et al., 2000) |
| Ventricular fibrillation | 0.91 | 1.90 | Patients developing VF in the setting of acute MI are at higher risk of in-hospital mortality. (Bougouin et al., 2014) |
| Third-degree AV block | 0.69 | 1.71 | In patients with CHB complicating STEMI, there was no change in risk-adjusted in-hospital mortality during the study period. (Harikrishnan et al., 2015) |
| Ventricular tachycardia | 0.51 | -0.19 | Ventricular tachycardia is most commonly associated with ischemic heart disease or other forms of structural heart disease that are associated with a risk of sudden death. (Koplan & Stevenson, 2009) |
| Dressler syndrome | 0.32 | -2.85 | The prognosis for patients with DS is typically considered to be quite good. (Leib et al., 2017) |
| Supraventricular tachycardia | 0.24 | -0.36 | Although SVT is usually not life-threatening, many patients suffer recurrent symptoms that have a major impact on their quality of life. (Medi et al., 2009) |
| Post-infarction angina | -1.40 | -2.85 | After adjustment, angina was only weakly associated with cardiovascular death, myocardial infarction, or stroke, but significantly associated with heart failure, cardiovascular hospitalization, and coronary revascularization. (Eisen et al., 2016) |

