# OpenReview forum: "Concept Gradient: Concept-based Interpretation Without Linear Assumption"
_ICLR.cc/2023/Conference — ICLR 2023 poster_

### Official Review · Reviewer_txu2 · 2022-10-20

**Confidence:** 3
**Correctness:** 3
**Technical Novelty And Significance:** 3
**Empirical Novelty And Significance:** 3
**Recommendation:** 6

**Clarity, Quality, Novelty And Reproducibility:**

### Clarity
The paper is relatively well written, although there are quite a few typos/articles missing which reduces the readability a bit and could be easily avoided.
For example
- in the abstract: and performed case study on a $ \rightarrow $ and performed a case study on a
- in 1: though there are an infinite number of functions exist $\rightarrow$ though there are an infinite number of functions
- in 3.4: we can directly constraint c within the $ \rightarrow $ we can directly constrain c within the
- in 3.4: Despite infinite number of choices $ \rightarrow $ Despite the/an infinite number of choices

Motivation and approach are clearly outlined.

### Quality
Theoretical motivation, derivations, and experiments are all of good quality.

### Novelty
The authors extend an already existing method and show its superior mathematical motivation and better experimental results.

### Reproducibility
Details for experiments are stated in the appendix but there is no code being published.

**Strength And Weaknesses:**

### Strengths
The idea is convincing and the theoretical motivation seems sound.
I do appreciate the toy examples that highlight where CAV might fail although it does not seem relevant in practice.
Experiments span different data sets and classifiers and a comparison to CAV shows significantly better performance of the proposed CG method, especially for global explanations.

### Weaknesses
The local concept recalls in figure 1 do not look that convincing for CAV or CG. Table 4 could include CAV values for comparison.




**Summary Of The Paper:**

The authors propose a novel concept-based explanation method: Concept Gradient (CG). In contrast to previous work, they use non-linear concept functions and show how the standard Concept Activation Vector (CAV) approach can be generalized to this setting.
Experimental results show superior performance of CG over CAV.

**Summary Of The Review:**

Overall I think the paper presents a nice contribution and I would lean toward publication at ICLR.

---

> ### Author Response · Authors · 2022-11-16
> **Response to Reviewer txu2**
>
> Dear reviewer,
>
> Thank you for the feedback and special attention to readability/typos, which we’ve fixed accordingly. As suggested we’ve also included CAV scores in Table 4 for comparison. CAV and CG interpretations are mostly aligned except for some concepts where CAV deviate from the literature descriptions. For instance, CG attributes high importance to “chronic heart failure”, a condition of about 40% mortality rate per year, while TCAV attributes extremely low importance. The inclusion of CAV alongside benefits understanding of the table, especially for readers unfamiliar with the medical domain.
>
> The inferior local concept recalls might be caused by the source of the ground truth label. The attribute labels of CUB are collected from MTurk, where individual contributors may not all be bird experts. Thus, individual ground truth labels may be quite noisy (also pointed out in [1]). This may explain the large discrepancy between global and local recall performance, since the ground truth labels for global recall are majority-voted.
>
> Finally, for reproducibility we will be publishing the code on Github to reproduce all the experiments in this paper as well as the implementation of CG after the review period ends. We've also uploaded the source code for all the experiments in the supplementary materiel.
>
> [1] Koh, Pang Wei, et al. "Concept bottleneck models." International Conference on Machine Learning. PMLR, 2020.

---

> > ### Comment · Reviewer_txu2 · 2022-11-18
> > **Thanks**
> >
> > Thanks for the clarifications. Best of luck for the publication.

---

### Official Review · Reviewer_UPnS · 2022-10-21

**Confidence:** 3
**Correctness:** 2
**Technical Novelty And Significance:** 3
**Empirical Novelty And Significance:** 3
**Recommendation:** 5

**Clarity, Quality, Novelty And Reproducibility:**

### Clarity
The paper is written clearly and the main idea is easy to follow.

### Quality, Novelty
The proposed score based on the inner product $\frac{\partial c}{\partial x}^\top \frac{\partial y}{\partial x}$ will be the main novelty of the paper.
However, its justification based on chain rule and pseudo-inverse is questionable.
To me, there is a more natural way to interpret the score as a (scaled) correlation between $f$ and $g$.

### Reproducibility
The authors provided the experimental details in Appendix.

**Strength And Weaknesses:**

### Strength
The proposed score $\frac{1}{\left\|\frac{\partial c}{\partial x}\right\|^2} \frac{\partial c}{\partial x}^\top \frac{\partial y}{\partial x}$ seems to be a reasonable one for quantifying the correlation between the model's outcome and the concept.
Essentially, $\frac{\partial c}{\partial x}^\top \frac{\partial y}{\partial x}$ is away from zero when both $y = f(x)$ and $c = g(x)$ changes simultaneously for moving $x \to x + \Delta x$ for a certain choice of $\Delta x$.
This is intuitive in a sense that moving $x$ to a direction changing $y$ also changes the strength of the concept $c$ simultaneously, indicating that $y$ and $c$ are correlated.

### Weakness
The essential weakness of the paper is on the justification of the score $\frac{1}{\left\|\frac{\partial c}{\partial x}\right\|^2} \frac{\partial c}{\partial x}^\top \frac{\partial y}{\partial x}$.
In the paper, the authors first adopted the sensitivity $\frac{\partial y}{\partial c}$ as the ideal ground truth, and then derived its estimate $\frac{1}{\left\|\frac{\partial c}{\partial x}\right\|^2} \frac{\partial c}{\partial x}^\top \frac{\partial y}{\partial x}$ by using pseudo-inverse.
This justification is questionable in two ways.
First, $\frac{\partial y}{\partial c}$ is not well-defined in general unless $g$ has inverse mapping.
Second, the choice of pseudo-inverse has a large degree of freedom.
In the paper, the authors adopted the pseudo-inverse with the minimum norm, i.e., Moore-Penrose inverse.
It is not very clear to me whether these questionable steps are required to justify the score $\frac{1}{\left\|\frac{\partial c}{\partial x}\right\|^2} \frac{\partial c}{\partial x}^\top \frac{\partial y}{\partial x}$.
Indeed, as the authors reported in the results, treating multiple concepts together leads to inferior performance.
This finding implies that the form of the inner product $\frac{\partial c}{\partial x}^\top \frac{\partial y}{\partial x}$ will be a more appropriate expression than $\left(\frac{\partial c}{\partial x}\right)^\dagger \frac{\partial y}{\partial x}$ obtained from these questionable steps.
As I pointed out in Strength, it would be more natural to interpret the score as a (scaled) correlation between $f$ and $g$.

**Summary Of The Paper:**

This paper proposes a concept-based explanation called Concept Gradient (CG), which is a non-linear extension of Concept Activation Vector (CAV).
The idea of CG is as follows.
Suppose the outcome of the model $f$ is given as $y = f(x)$.
Similarity, there is another model $g$ predicting the concepts $c = g(x)$.
The authors then proposed to use the sensitivity of the outcome with respect to the concept change $\frac{\partial y}{\partial c}$ as the contribution of the concept $c$ to the model's outcome $y$ for the input $x$.
To compute $\frac{\partial y}{\partial c}$, the authors introduced chain rule $\frac{\partial y}{\partial c} = \frac{\partial x}{\partial c} \frac{\partial y}{\partial x} \approx \left(\frac{\partial c}{\partial x}\right)^\dagger \frac{\partial y}{\partial x}$ where $\dagger$ denotes pseudo-inverse.
In particular, the authors reported that treating each individual concept separately induced better results.
In such a case, we have $\frac{\partial y}{\partial c} \approx \frac{1}{\left\|\frac{\partial c}{\partial x}\right\|^2} \frac{\partial c}{\partial x}^\top \frac{\partial y}{\partial x}$.
In the experiments, the authors reported that CG is better at identifying true relevant concepts compared to CAV.

**Summary Of The Review:**

I think the proposed score based on the inner product $\frac{\partial c}{\partial x}^\top \frac{\partial y}{\partial x}$ is novel and will be effective in practice.
However, its justification is questionable.
There may be some other ways to justify the score.

---

> ### Author Response · Authors · 2022-11-16
> **Response to Reviewer UPnS**
>
> Dear reviewer,
>
> Thank you for the critical comment on the motivation for CG and appreciation of the effectiveness of our proposed method. We agree that viewing CG as a form of gradient similarity measurement between the target and concept model is a good viewpoint for understanding CG. In fact, this view is also stated in the paragraph right after Eq(4). The only difference comes down to scaling. Although the scaling may not affect the results in some practical cases, in theory (and perhaps other use cases) the correct scaling $\frac{1}{\|\nabla g(x)\|^2}$ is important. Simply taking the normalized gradient inner product would not lead to the correct scaling.
>
> To demonstrate why the scaling term derived by CG is ideal, we discuss cases where $\nabla g(x)$ is invertible or not. In the non-invertible case we will also explain why we choose the minimum-norm solution for pseudo-inverse.
>
> > When $\nabla g(x)$ is invertible:
>
> Taking the pseudo-inverse exactly recovers the inverse matrix. CG is equivalent to the actual gradient $\frac{\text{d}y}{\text{d}c}$, as demonstrated in Eq(3). CG won’t recover $\frac{\text{d}y}{\text{d}c}$ with any other scaling factor or scaling matrix.
>
> > When $\nabla g(x)$ is non-invertible:
>
> 1. Why should we choose a minimum-norm solution for pseudo-inverse?
>
>     We’ve included the below clarification for the justification of using the pseudo-inverse in Appendix C. Please refer to the Appendix for better formatting.
>
>     Let’s first understand the issue with a concrete version of the example sketched in Section 3.4:
>     * The raw input x has two dimensions: $x_1$ indicates the feature “color” and $x_2$ indicates the feature “shape”.
>     * There’s only one single concept $c$ corresponds to “color”, so $c = g(x) = x_1$
>     * The target model is $y=f(x) = x_1 - x_2$.
>     * We consider a particular input $\hat{x} = [1, 0]$ and $\hat{c}=1$.
>
>     In this example, clearly a small perturbation to “color” can lead to linear change to output with slope 1, so the concept attribution score should be 1.
>
>     Now let’s follow the derivation of CG in this example to show why we choose the minimum-norm solution. Centered at $\hat{c}$, there can be infinite number of (linear) mappings from $c$ to $x$, but all of them will follow the form
>
>     $h(c) = h(\hat{c}) + \nabla h(\hat{c}) (c-\hat{c})$  and
>     $\nabla h(\hat{c}) = \nabla g(\hat{x})^\dagger + G_\perp$ (according to Theorem 1)
>
>
>     In our example, $h$ can be written explicitly as follows:
>
>     $$h(c) = [1, 0] + ( [1, 0] + [0, \alpha]) (c-1) = [c, \alpha (c-1) ], \quad \forall \alpha$$
>
>
>     This corresponds to Theorem 1, where $\nabla g(\hat{x})^+ = [1,0] $ and $G_\perp$ is the set of $[0, \alpha]$ with any $\alpha$. These functions all set $x_1=c$ but will set $x_2=\alpha (c-1)$ with arbitrary $\alpha$. Conceptually, this is saying there are infinite number of ways to form the mapping from “color” ($c$) to input ($x$): we always change “color” according to $c$ but we can arbitrarily change the other orthogonal feature “shape” ($x_2$) according to color. Concept Gradient is trying to see how the change of $c$ affect output $y$ while **keeping all the other orthogonal factors fixed**. Therefore, we want to find the mapping that has minimal perturbation to $x$ when changing $c$, corresponding to keeping all other orthogonal factors fixed. This is the reason to choose the minimal norm solution in pseudo inverse.
>
>     In this example, our method sets $\alpha=0$ and will lead to $h(c) = [1, 0]$. If $\alpha$ is set to some non-zero values in the pseudo-inverse, it will lead to the following score:
>
>     $\frac{\partial f(h(\hat{c}))}{\partial c} =  \langle [1, \alpha], [1, -1] \rangle = 1-\alpha$, which will give the wrong attribution score for any $\alpha\neq 0$.
>
>     We’ve included the above clarification for justification for using the pseudo-inverse in Appendix C.
>
> 2. Is the scaling factor important?
>
>     Having the correct scaling factor may be important, for instance when the value in each dimension of the input representation varies significantly (e.g. models without BatchNorm). In the simple 2-dimensional toy example in Eq(5), CG (with correct scaling) yields the correct attribution score while using the scaling factor of $\frac{1}{\|\nabla g(x)\|}$ leads to error and incorrect ranking of concept importance. Further, if no scaling factor is applied the gradient inner product measures the contribution of $c_0$ as 1000 and the contribution of $c_1$ as 1, which is even worse than using $\frac{1}{\|\nabla g(x)\|}$. This demonstrates the scaling factor could be important.
>
> We hope that our discussion above clarified our motivation for CG and provided an explanation for why viewing CG as a normalized gradient inner product, albeit seemingly more straightforward, may not be more “natural”. This is an important finding in this paper, and we think the scaling factor issue, although subtle, is interesting to be introduced to the community.

---

> > ### Comment · Reviewer_UPnS · 2022-11-28
> > **Thank you for the reply**
> >
> > I would like to thank the author for the detailed reply.
> >
> > Here, I would like to describe my question in detail.
> > We can view the proposed CG as the solution to the following least squares regression (LSR):
> > $$
> > \beta = \arg\min\_\beta \left\\| \frac{\partial y}{\partial x} - \frac{\partial c}{\partial x} \beta \right\\|^2
> > $$
> > For the case when $\frac{\partial c}{\partial x}$ is invertible, we have $\beta = \frac{\partial c}{\partial x}^{-1} \frac{\partial y}{\partial x}$.
> > For the case when $\frac{\partial c}{\partial x}$ is not invertible, we can adopt arbitrary methods for least squares regression such as reguralization and the minimum norm solution.
> >
> > When we view CG from the perspective of LSR, we can find that CG inherits the weakness of LSR.
> > It is known that the solution of LSR is sensitive to the correlation between the "features", which corresponds to the correlations between the concepts in the gradient $\frac{\partial c}{\partial x}$.
> > If there are multiple concepts that share similar gradients, their corresponding regression coefficients $\beta$ (i.e., CG) can be almost arbitrary.
> > For example, if we inject small noises to such gradients, there is a possibility that the winning concept having the large $\beta$ among correlated concepts can change drastically.
> > From this observation, I considered that defining CG based on LSR can be ill-posed and will not be appropriate.
> >
> > By contrast, if we define CG as the (possibly scaled) dot-product directly without LSR, we can avoid the above trouble coming from correlated concepts.
> > This correponds to measureing CG as the (scaled) correlation between the gradients, rather than the (scaled) conditional correlation in LSR.
> > When I view CG from this perspective, the current paper looks that correlation and conditional correlation, the different terminologies in statistics, are completely mixed up.
> >
> > In summary, here I would like to restate my question, what is the benefit of defining CG as (scaled) conditional correlation rather than correlation, even though in most parts of the papers CG is considered as correlation rather than conditional correlation?

---

> > > ### Author Response · Authors · 2022-11-30
> > > **Response to Reviewer UPnS**
> > >
> > > We sincerely thank the reviewer for analyzing the derivation of CG with attention to technical details. We completely agree that when the gradients for concepts are similar (or even duplicates), perturbations on the gradients impact CG importance attribution. This is why we proposed two methods for calculating the pseudo-inverse when evaluating CG — **individually** and **jointly**. These two approaches correspond to two different types of interpretations (as detailed in Section 3.3). Adopting the reviewer’s terminology, CG with **individual** pseudo-inverse corresponds to “correlation”, while CG with **joint** pseudo-inverse corresponds to “conditional correlation”.
> > >
> > > To answer the reviewer’s question, we are not suggesting that CG should be evaluated **individually** (correlation) or **jointly** (condition correlation), since that is up to the user's choice. Both versions of CG can be derived from gradient chain rule, as opposed to gradient inner product which leads to incorrect scaling. We show our example demonstrating how (scaled) gradient inner product leads to incorrect scaling while CG is correct (Section 3.2) in a concise form in the image link below.
> > >
> > > https://imgur.com/a1sWU6L
> > >
> > > In fact, it’s easy to verify that CG will correctly obtain the gradient no matter which layer is chosen for computation. In conclusion, CG with **individual** pseudo-inverse obtains the correct scaling while *not* suffering from the LSR sensitivity issue.
> > >
> > > We genuinely appreciated the reviewer helping us clarify the derivation of CG and hoped that the above discussion resolved the reviewer’s concern. Please let us know if there are any other issues we could further discuss and try to clarify. Thank you.

---

> > > > ### Comment · Reviewer_UPnS · 2022-12-08
> > > > **Reply after discussion with other reviewers**
> > > >
> > > > I had some discussions with other reviewers.
> > > > Some of the reviewers agreed with that (i) the individual CG will be beneficial, and (ii) the derivation of the individual CG as a special case of the joint CG based on chain rule can be too convoluted.
> > > >
> > > > For (i), I am convinced with the practical utlity of the individual CG but not the joint CG.
> > > > I agree that this finding would be valuable to the community.
> > > >
> > > > My concern is all about (ii).
> > > > The reason that I cannot be positive towards this paper is that this finding is derived from a possibly inappropriate way.
> > > > As I repeat several times, I think it would be more natural to define the individual CG as CG directly.
> > > > I am happy to raise my score if this point could be resolved.
> > > >
> > > > In the current state, I have to say that I am not fully convinced.
> > > > Although the authors raised a synthetic example that the individual CG using psuedo-inverse provides a correct scaling, the generality of this claim is questionable.
> > > > Below, I show that something wierd can happen.
> > > >
> > > > Let
> > > > * $y = h\_0$
> > > > * $c\_0 = a h\_0 + (1 - a) h\_1$, $a \in [0, 1]$
> > > > * $c\_1 = b h\_0 + (1 - b) h\_1$, $b \in [0, 1]$
> > > >
> > > > In this case, $y = c_0$ holds when $a = 1$.
> > > > Hence, it is expected that the contribution of $c_0$ to $y$ is maximized at $a=1$.
> > > > However, the propsoed individual CG provides
> > > > $\frac{\frac{dy}{dh}^\top \frac{dc\_0}{dh}}{\left\\| \frac{dc\_0}{dh} \right\\|^2}=\frac{[1, 0][a, 1-a]^\top}{a^2 + (1 - a)^2} = \frac{a}{a^2 + (1 - a)^2}$.
> > > > This score gets 1 when $a = 1$, while $a = \frac{1}{\sqrt{2}}$ provides the score $\frac{1}{2 \sqrt{2} - 2} > 1$.
> > > > That is, $a = 1$ does not provide the maximum.
> > > >
> > > > This example provides a counter-evidence to the claim, and the scaling provided by the proposed CG may not be ideal.

---

> > > > > ### Author Response · Authors · 2022-12-08
> > > > > **Response to Reviewer UPnS**
> > > > >
> > > > > Dear Reviewer,
> > > > >
> > > > > We are happy to reach consensus on the practical effectiveness of CG. Furthermore, we also agree that it is rather convoluted and potentially deters readers by first defining the general framework of CG then introducing individual CG as a special case. CG is *not that complicated* in practice and we do wish to present it in the easiest way to possibly understand (without losing detail). Thus, our newly revised Section 3, if desired by the reviewers, can be structured as follows:
> > > > >
> > > > > 1. Introduce individual CG (original Section 3.2)
> > > > >     * Present the explicit form of scaled inner product and intuition
> > > > >     * Clarify the importance of correct scaling
> > > > > 2. Extend individual CG to joint CG for ideal theoretical analysis (original Section 3.1)
> > > > > 3. Describe the difference between the two approaches (original Section 3.3)
> > > > > 4. Derive CG, which is valid for both individual and joint (original Section 3.4)
> > > > > 5. Implementation of CG (original Section 3.5)
> > > > >
> > > > > **Essentially we are proposing to swap the order of Section 3.2 and 3.1 to provide readers a straightforward and gentle introduction to CG.**
> > > > >
> > > > >
> > > > > As for the example presented by the reviewer, contrary to the reviewer’s hypothesis, we shouldn’t expect the contribution of $c_0$ to be maximized when $a = 1$, if considering how $\epsilon$ change in the concept affects the target output (which is the core assumption of gradient-based explanations). To gain a better intuition, let’s consider an even simpler scenario where $y = h_0$ and $c_0 = a \cdot h_0$, removing the dependency of $c_0$ on $h_1$. In this case, $y = \frac{1}{a} \cdot c_0$ and the smaller $a$ is, the more relevant $c_0$ is to $y$. Thus, this does not serve as a counter-example to CG scaling. Nevertheless, it is an interesting artificial scenario to study in a future work.
> > > > >
> > > > > To conclude our response, we would like to thank the reviewer once again for firmly  demanding the paper to be as easy to understand, which we believe benefits all readers. We are open to making the corresponding changes in Section 3 (re-ordering the paragraphs) and expect to update the new manuscript in the camera-ready version.

---

> > > ### Author Response · Authors · 2022-12-07
> > > **Followup on author response to Reviewer UPnS**
> > >
> > > Dear reviewer,
> > >
> > > We would like to answer potential follow up questions of yours given our response. In our reply we emphasized the importance of deriving CG from a gradient chain rule standpoint — the scaling factor. As tempting as it might be to reduce CG to a simpler form of gradient inner product, we provided a concrete example where only by applying the correct scaling (in CG) would lead to the correct attribution in Section 3.2. We’ve also elaborated on the choice of selecting the Moore-Penrose inverse for inverting the concept gradient matrix in Appendix C. We will go over example once again:
> > >
> > > Consider $f$ as the following network with two-dimensional input $[x_0 , x_1 ]$:
> > >
> > > $$
> > > y = 0.1z_0 + z_1, \quad z_0 = 100 h_0, \quad z_1 = h_1, \quad h_0 = 0.01 x_0, \quad h_1 = x_1
> > > $$
> > >
> > > Consider concepts $c_0 = x_0$ and $c_1 = x_1$. Then we know since $y=0.1z_0+z_1 =0.1x_0+x_1$, the contribution of $c_1$ should be 10 times larger than $c_0$. In fact, $dy/dc_0 = 0.1, dy/dc_1 = 1$ and it’s easy to verify that CG will correctly obtain the gradient no matter which layer is chosen for computing. However, the results will be incorrect when a different normalization term is used when computing concept explanation on the hidden layer $h = [h_0, h_1]$. Since $c_0 = 100h_0$, $c_1 = h_1$, $y = 10h_0 + h_1$, we have
> > >
> > > $$
> > > \text{For } c_0: v = dc_0/dh = [100,0]^T , u = dy/dh = [10,1], v^T u/\|v\| = 10,v^T u/\|v\|2 = 0.1
> > > $$
> > > $$
> > > \text{For } c_1: v = dc_1/dh = [0,1]^T , u = dy/dh = [10,1], v^T u/\|v\| = 1,v^T u/\|v\|2 = 1
> > > $$
> > > Therefore, the normalization term used with **gradient inner product** ($v^T u / \|v\|$) will lead to a conclusion that $c_0$ is more important than $c_1$, while **CG with chain rule** ($v^T u / \|v\|^2$) will correctly get the actual gradient and conclude otherwise. CG attributes correctly since it is formally derived from the actual gradient. In contrast, basing the attribution on the intuition of “gradient sensitivity” and not the exact chain-rule gradient subjects to (incorrect) per-dimension scaling.
> > >
> > > We agree with the reviewer that the success of CG is achieved by capturing the relation between the sensitivity of target and concept models, for which deriving the “concept gradient” with chain rule accomplishes. The intuition behind CG was concisely summarized in the reviewer’s feedback, which leads us to believe that *we do have consensus* regarding the efficacy and reasonableness of CG in the broader sense. Please let us know if there are any other issues we could further discuss and try to clarify. Thank you.

---

### Official Review · Reviewer_zoZR · 2022-10-25

**Confidence:** 4
**Correctness:** 3
**Technical Novelty And Significance:** 4
**Empirical Novelty And Significance:** 3
**Recommendation:** 8

**Clarity, Quality, Novelty And Reproducibility:**

The work is clearly written, with well explained concepts and clarifying examples.

There is a minor inaccuracy in the notation, as at the end of page 3, Vc represents both g(x) and its gradient.

**Strength And Weaknesses:**

PROs

- The work addresses a clear limitation of the CAV approach, that severely affects its practical applicability.

- The derivation is sound

_ The experimental evaluation is extensive and convincingly supports the advantage of the approach wrt to CAV.

CONs

- A critique one could make to this line of research is that if
  concepts are available, one should aim at directly enforcing models
  to use them (the concept-based learning models) rather than testing
  their alignment post-hoc.

- The complete version of the approach, jointly computing concept
  gradient for all concepts, seems to provide worse results in terms
  of alignment with concept importance as assessed by humans. To me
  this again hints at a limitation of this research line (see the
  summary of the review).

**Summary Of The Paper:**

The work extends the CAV approach for concept-based explainability to
deal with concepts that cannot be expressed as linear combinations of
input (or latent) features.


**Summary Of The Review:**

Overall, I think this is a reasonable contribution to the XAI field,
fixing a clear deficiency of existing solutions for concept-based
post-hoc explanations.  My main concern is on the limitations of
concept-based post-hoc explanation itself. However I do not think this
is a reason to disregard this research line and this work in
particular.  Rather, I would like the authors to discuss in more
detail the pros and cons of post-hoc interpretation with respect to
concept-based models. The authors did briefly mention this in the
related work section, but I think a deeper analysis is needed. For
instance, the suboptimal performance of concept gradient computed
jointly over concepts hints at the lack of orthogonality /
disentanglement between "concepts" learned by the network, something
that the research on concept-based models is actively addressing.

---

> ### Author Response · Authors · 2022-11-16
> **Response to Reviewer zoZR**
>
> Dear reviewer,
>
> Thank you for your feedback and comment on concept-based post-hoc explanations. We completely agree that the model may be more “explainable” if directly using pre-defined concepts in the first hand. However, adding this restriction may degrade the model’s performance, which is not acceptable in many applications. Therefore, being able to interpret with respect to arbitrary concepts after a model is trained remains practically useful, as the reviewer also agreed. We sincerely appreciate the reviewer’s recognition of progression in all directions of XAI. Nevertheless, we’ve revised the related work section suggesting users to adopt concept-based models if the use case allows.
>
> One last note: to slightly clarify the mentioned joint pseudo-inverse over concept example, the suboptimal is likely because similar concepts exist in the set of concepts used for interpretation. Joint pseudo-inverse interprets the concepts in a “what happens if we remove one concept while keeping all other concepts” fashion, where a concept  would be attributed low importance if it is replaceable by other concepts. On the other hand, performing pseudo-inverse independently for each concept disregards other concepts, which is more similar to human interpretation. We agree that this indicates in practice many concepts are correlated, and how to properly disentangle the concepts is an important research direction.

---

> ### Author Response · Authors · 2022-11-28
> **Followup on author response to Reviewer zoZR**
>
> Dear reviewer,
>
> As the reviewing deadline approaches, we would like to answer potential follow up questions of yours given our response. To recap, we agreed the importance of adopting concept-based models if possible and acknowledged its importance in the main paper (related works section). As concept-based models without drawbacks and limitations are being developed, post-hoc explanations like CG fills the gap of bridging interpretability and complex neural network models. It is an important field of study, as mentioned in the reviewer’s feedback, in terms of its practicality and understanding of neural networks, providing guidelines for improving concept-based models. We appreciate the reviewer’s affirmative perspective regarding CG for its derivation soundness and advantage over CAV. Please let us know if there are any other issues we could further discuss and try to clarify. Thank you.

---

> > ### Comment · Reviewer_zoZR · 2022-12-08
> > **Response to follow-up**
> >
> > Dear authors,
> > I would like to thank you for clarifying the limitations and role of post-hoc interpretation as compared to concept-based models. My impression is that the limitations that we see in the results are intrinsic in the lack of disentanglement  in the learned concepts, something which is independent of the proposed approach and that suggests that further research is needed in that direction. Overall I am convinced by the way the approach is presented and contextualized in this version of the manuscript.

---

### Official Review · Reviewer_pw8j · 2022-10-26

**Confidence:** 4
**Correctness:** 3
**Technical Novelty And Significance:** 3
**Empirical Novelty And Significance:** 2
**Recommendation:** 6

**Clarity, Quality, Novelty And Reproducibility:**

For the most part, the methodology is clear.
The provided formulations about gradient of class with respect to concept is insightful and novel.
I have some concerns that I hope the authors can address.

**Strength And Weaknesses:**

+The method is simple.
+The provided formulations about gradient of class with respect to concept are insightful. Although, eventually it all reduces to a simple dot product of two gradient vectors.


-Why finetuning the same network for g? Specially for deeper layers, g becomes very shallow and simple. Why should one not use a more complex model for g?

-Concept accuracy in Table 1 and 2 are very close suggesting that a linear model is not as bad as one may think (0.72 vs 0.79 in Res50). Which layer is used for generating the results in Table 1 and 2?

-TCAV does not simply use directional derivative which is analogous to the proposed local CG. Instead, they define TCAV_Q as “the fraction of k-class inputs whose l-layer activation vector was positively influenced by concept C”. In contrast the proposed global CG is defined based on sum of sign(local CG). What is the advantage of this definition over the definition of TCAV_Q score?

-TCAV performs a statistical significance testing and uses only the results that pass this test (Kim 2018, section 3.5). Did you do this step?

-When comparing to TCAV, did you rank the concepts according to TCAV_Q score from “relative” CAV (Kim 2018, section 3.6) which is the recommended way of using the method when assessing relative importance of multiple concepts? Please clarify these details in the paper.

-Simply using gradient to measure feature importance is widely admitted to be misleading and that’s why many gradient-based methods have been trying to generating more faithful saliency maps (Adebayo 2018). CG score is purely based on gradient. Does CG suffer from similar issues?
 See Adebayo , et al Sanity Checks for Saliency Maps, Neurips 2018.

-In the experiments on the third dataset about mortality risk, have you had the results reviewed by a medical expert? Since I, and likely other reviewers, are unable to comment on the validity of these results, I was hoping you could provide TCAV_Q scores too. Having both CG and TCAV_Q scores side by side would enable us to at least see if there are any disagreement between the two method and if CG has any advantage over TCAV_Q on this dataset.

- CG is defined to measure how small perturbations on each concept affect the label prediction. CG essentially is dot product of the two gradient vector (after a normalization which is shown not to be important). The dot product is a symmetric operation. So it also measures how small perturbations on a predicted class affect the concept prediction. Right?

Typo: Pp 15 Appendix D para. 2 Fig ??


**Summary Of The Paper:**

The paper propose a method for explaining intermediate activations of a neural network using concepts. In a nutshell, f(x) and g(x) are the networks that take activations x from an intermediate layer as input and respectively predict class and concept. The relevance of a concept c to a class y is measured using the proposed Concept Gradient (CG) score, which is based on the gradient of f and g with respect to x. CG score is calculated as normalized dot product of the two gradients. The proposed method is compared with TCAV (Kim 2018) on two datasets (CUB and AWA). The proposed method is also applied to a third dataset but the results are not compared with TCAV or any other method.

**Summary Of The Review:**

I have some questions that I hope the authors can clarify.

---

> ### Author Response · Authors · 2022-11-16
> **Response to Reviewer pw8j (1/2)**
>
> Dear reviewer,
> Thank you for the detailed review and extensive questions. We will answer each question individually in the following section:
>
> > Why finetuning the same network for g? Especially for deeper layers, g becomes very shallow and simple. Why should one not use a more complex model for g?
>
> Our original hypothesis is that input representation in deeper layers contain higher level semantics, which requires less expressiveness to model. Thus, finetuning g is sufficient. To test this hypothesis, we conducted an experiment which is added to Appendix G. We selected an input representation from a deeper layer (layer4.1 and layer4.2 from ResNet50) and compared our original finetuned model with more complex variants. The results show that our hypothesis holds and that finetuning the same model is sufficiently expressive.
>
> In general, concept models can be arbitrary and need not to be finetuned from the target model f in order for CG to work. It is possible in some cases where finetuning the same network is insufficient to capture the concept. This is highly dependent on the use case and we would advise users to select suitable function classes to represent their concepts.
>
> > Which layer is used for generating the results in Table 1 and 2?
>
> The layer is selected via the best recall for each individual method. For Table 1, CG is evaluated on `Mixed_7c`, `layer4.2`, and `classifier.3` for the three models in order while CAV is evaluated on the final layer. For Table 2, CG is evaluated on `Mixed_7a`, `layer4.2`, and `classifier.3` for the three models in order while CAV is evaluated on `Mixed_7b`, `layer4.1`, and `classifier.0`. The reason we report the best layer and not just select a layer to compare is to give CAV an edge. As shown in Fig 1, the performance of CG is consistent across layers (consistently good) while CAV is sensitive to layer selection. Comparing the same layer puts CAV at a disadvantage.
>
> > Why is Global CG defined differently from TCAV_Q?
>
> That is unfortunately a blunder on our part. The global CG is in fact defined exactly as the TCAV_Q score: the portion of positive local CGs. It is simply written incorrectly in the paper. We’ve fixed the definitions. To answer the original question, no, there is no benefit in taking the sign instead since it’s just a linearly scaled and shifted version of the TCAV_Q definition.
>
> > Did you perform T-test for TCAV?
>
> Yes.
>
> > Why not use “relative” CAV to assess relative importance of multiple concepts?
>
> To the best of our understanding, the relative CAV score is used to distinguish between two concepts, since a hyperplane formed by the relative concept vector can only partition the space into two. Thus, it is not applicable to interpretations wrt 2+ concepts.

---

> ### Author Response · Authors · 2022-11-16
> **Response to Reviewer pw8j (2/2)**
>
> > Does CG suffer from unfaithful gradient issues?
>
> It is true that input gradients may not be robust in highly non-linear models. In standard feature-based attribution, many methods such as SmoothGrad [1] and Integrated Gradients [2] can improve faithfulness. To examine whether CG suffers from the same issues, we implemented SmoothCG (analogue to Smooth Gradient) — a variant of CG that smoothes over the neighborhood near the data sample, which is less prone to sensitivity to non-linearity, and compared it with the original CG (see details in Appendix H).
>
> In conclusion, SmoothCG performed largely the same as CG. SmoothCG is marginally better when interpreting in shallow layers of a neural network, likely due to the less accurate gradients, but slightly worse in deeper layers when gradients are more accurate. Given the marginal difference, we conclude that empirically CG works sufficiently well. That being said, if the gradients for the particular use case are known to be inaccurate, adoption of SmoothCG is recommended.
>
> [1] Smilkov et al., 2017. SmoothGrad: removing noise by adding noise.
>
> [2] Sundararajan et al., 2017. Axiomatic Attribution for Deep Networks.
>
> > Include CAV for Myocardial Infarction Complication dataset for comparison. Are results reviewed by a medical expert?
>
> The provided medical literature reference was intended to serve as the expert guidance. Unfortunately we haven’t been able to consult a medical professional yet. As suggested by the comment, we included the TCAV_Q scores alongside CG for reference. We edited Section 4.3 to compare between the two interpretation methods. The interpretations are mostly aligned with some exceptions of TCAV deviating from medical literature. For instance, CG attributes high importance to “chronic heart failure”, a condition of about 40% mortality rate per year, while TCAV attributes extremely low importance.
>
> > CG is essentially  dot product of the two gradient vectors and thus measures how small perturbations on a predicted class affect the concept prediction?
>
> CG is close to the normalized gradient dot product but not quite (as demonstrated in Section 3.2). However, empirically it is shown that the neural network models (at least in our experiments) are well-behaved and the scaling is not as relevant. So theoretically CG does not measure how small perturbations on a predicted class affect the concept prediction but empirically it might be a good indicator.
>
> We hope that our added experiments in the appendix and elaborations above answered all your questions. Please let us know if there is anything else that might be of your concern. We will be more than happy to provide additional clarification!

---

> ### Author Response · Authors · 2022-11-28
> **Followup on author response to Reviewer pw8j**
>
> Dear reviewer,
>
> As the reviewing deadline approaches, we would like to answer potential follow up questions of yours given our response. To recap, we clarified technical implementation details of CG and its comparison with CAV and fixed the typos for better readability. For the concern regarding faithfulness of gradients, we conducted additional experiments on a smoothed version of CG (SmoothCG) where the results are expected to be less sensitive to gradients. Comparison between the variants show that the impact of gradient faithfulness is minimal on CG.
>
> CG is a simple and insightful method as agreed upon by the reviewer, and we’ve demonstrated its efficacy with empirical studies. Our additional experiments and ablation studies tackle the edge cases of CG usage. The results suggest that CG is effective in various settings (e.g. tasks, model architectures, layers), supporting it as a general method to perform concept-based interpretation. Please let us know if there are any other issues we could further discuss and try to clarify. Thank you.

---

### Decision · Program_Chairs · 2023-01-20

**Decision:**

Accept: poster

**Justification For Why Not Higher Score:**

As suggested by the reviewers, the paper builds on recent work and improves upon concept-based methods. While the community may benefit from seeing this work and further building on it, and the empirical results are convincing, the paper also does not present any major breakthroughs that would need to be further highlighted. Further, presentation clarity needs significant improvements, as discussed in the weaknesses section. In particular, given the comments of Reviewer UPnS, it seems like CG and joint CG could be introduced in a much clearer way, explaining why the scores may or may not be susceptible to changes in scaling of the dependence among the outputs and concepts.

The authors agreed to make the following additional edits after further discussion with the reviewers:
- “Essentially we are proposing to swap the order of Section 3.2 and 3.1 to provide readers a straightforward and gentle introduction to CG.”
- release the code.

**Justification For Why Not Lower Score:**

The proposed gradient-based method for identifying which concepts explain model's predictions is novel, and may be of interest to the community. The experiments were done on a variety of datasets, and the results indicate significant improvements over CAV.

**Metareview: Summary, Strengths And Weaknesses:**

The proposed paper considers a setting where one wants to explain a model's predictions via some set of concepts. The authors quantify how relevant these concepts are via computing sensitivity of the model's predictions to changes in the concepts. In addition to the derivation of the score, some empirical results are also presented. The authors ran several more experiments analyzing potential pitfalls of their gradient-proposed method, as suggested by Reviewer pw8j.

One of the main weaknesses of the paper is the presentation of individual vs joint CG. It seems like it caused some confusion among the reviewers. I encourage the authors to implement proposed changes to the introduction of these concepts.

**Note From Pc:**

if the above contains the word "oral" or "spotlight" please see: "oral" presentation means -> notable-top-5% and "spotlight" means -> notable-top-25%. As stated in our emails, we are disassociating presentation type from AC recommendations